# LARO: Learning to Accelerate Two-Stage Adaptive Robust Optimization with Relaxation Guarantees

## Abstract

Two-stage Adaptive Robust Optimization (ARO) with discrete and polyhedral uncertainty sets incorporates "wait-and-see" decisions to reduce conservatism but remains intractable due to its multi-level structure and mixed-integer recourse. This paper introduces *LARO*, a learning-accelerated two-phase decomposition framework that scales ARO efficiently without embedding neural networks (NNs) into optimization models. The framework operates in two phases: a *Relaxed Master Problem* (RMP) that identifies candidate here-and-now decisions through a *penalized selection mechanism*, where pre-computed severity scores bias scenario choice toward adversarial cases, and a verification phase that ensures restricted worst-case consistency. By decoupling the NN from the RMP, we eliminate solver-compatible embeddings, reduce computational overhead, and enable the use of more expressive neural architectures for recourse evaluation in the adversarial step.

We establish finite convergence, with the number of iterations bounded by the size of the discrete uncertainty set, and show that the penalized RMP preserves valid lower bound (LB) while improving iteration efficiency by prioritizing impactful scenarios. Experiments on robust knapsack and unit commitment (UC) problems in power grids demonstrate the scalability of the framework, achieving runtime speedups of up to $10^3\times$ for knapsack instances and $10^2\times$ for a 24-bus power network compared to classical column-and-constraint generation. The solve spped is achieved while maintaining optimality gaps typically below 7% for knapsack instances and 2% on the UC problems. This work delivers a *severity-aware, learning-accelerated CCG* that is both scalable and certifiable, advancing robust decision-making under uncertainty.

## 1 Introduction

Robust Optimization (RO) is a fundamental framework for decision-making under data uncertainty, ensuring solutions remain feasible across a defined uncertainty set Bertsimas et al. (2011); Ben-Tal et al. (2009). Unlike deterministic or stochastic optimization, RO prioritizes reliability, making it valuable in finance, supply chains, and power systems. However, its conservatism in accounting for all uncertainties can reduce flexibility and increase costs.

Adaptive Robust Optimization (ARO), as formalized in Ben-Tal et al. (2004); Yanıkoğlu et al. (2019), extends the RO framework by incorporating *adaptive* decisions to mitigate excessive conservatism. ARO partitions decisions into "here-and-now" ($\mathbf{x}$) and "wait-and-see" ($\mathbf{y_\xi}$), where the latter responds non-anticipatively to uncertainty realizations $\boldsymbol{\xi} \in \Xi$. This adaptivity reduces the conservatism of static RO methods. We consider the following general ARO problem:

$$\min_{\boldsymbol{x} \in X} \max_{\boldsymbol{\xi} \in \Xi} \min_{\boldsymbol{y_\xi} \in Y(\boldsymbol{x}, \boldsymbol{\xi})} \quad \boldsymbol{c}^\top \boldsymbol{x} + \boldsymbol{d}^\top \boldsymbol{y_\xi}$$
$$\text{s.t.} \qquad \boldsymbol{T}(\boldsymbol{\xi})\, \boldsymbol{x} + \boldsymbol{W}\, \boldsymbol{y_\xi} \le h(\boldsymbol{\xi}). \tag{1}$$

The objective function in equation 1 minimizes the worst-case total cost over a fixed polyhedral uncertainty set $\Xi$; the recourse $\boldsymbol{y_\xi}$ adapts to $\boldsymbol{\xi}$ while remaining feasible.

Despite its theoretical appeal, the scalability of ARO presented in Equation 1 is fundamentally limited by its nested min-max-min structure. Ensuring that adaptive decisions $y_\xi$ are feasible for every possible realization of uncertainty $\xi \in \Xi$ creates a semi-infinite optimization problem. Analytically reformulating these infinite constraints as an adversarial problem introduces bilinear terms between decision variables and uncertain parameters. This results in a nonconvex and often non-differentiable second-stage value function, which invalidates the assumptions of classical convex decomposition algorithms like Benders' method Ben-Tal et al. (2009); Bertsimas et al. (2011).

The presence of mixed-integer variables compounds this difficulty, creating formidable mixed-integer nonconvex programs, a class of problems known to be NP-hard even in much static robust optimization with polyhedral uncertainty sets Bertsimas & Sim (2003). A common assumption in this paper, also utilized by Bertsimas & Kim (2024) and Dumouchelle et al. (2023) discretizes the uncertainty set $\Xi$, but the core issue still remains. It merely transforms the problem into a large-scale deterministic equivalent where the bilinear linkages persist, coupling decisions across numerous scenarios. Consequently, despite relaxing optimality guarantees, scalable ARO methods remain elusive; this work addresses that gap.

## 1.1 RELATED LITERATURE

Researchers have proposed methods to mitigate the issues associated with the computational complexity and infinite-dimensional nature of ARO. The traditional methods generally fall into two broad categories: restricting the wait-and-see decision thereby creating decision rules (Ben-Tal et al. (2004); Chen & Zhang (2009); Georghiou et al. (2021); Kuhn et al. (2011); Subramanyam et al. (2020) and decomposition-based algorithms Bertsimas et al. (2012), Zeng & Zhao (2013)). Restricting decision rules limits the flexibility of the wait-and-see (recourse) variables to often a lower-dimensional function class— affine or piecewise-affine policies Bertsimas & Goyal (2012). This can transform the ARO problem into a robust or convex optimization problem, reducing its complexity. However, such overly simplistic policy classes may lead to suboptimal or even infeasible solutions . Techniques such as Benders Decomposition Thiele et al. (2009) and Column-and-Constraint Generation (CCG) Zeng & Zhao (2013) aim for exact or near-optimal solutions by iteratively refining the set of constraints or variables. However, iteratively refining the feasible region and/or adding scenarios can lead to a large number of subproblems, making these methods intractable for large-scale applications. Moreover, in complex ARO problems (with mixed-integer recourse), the iterative process may require many iterations to reach near-optimal solutions.

More recently, machine learning (ML) techniques have emerged as a powerful alternative for tackling ARO problems in real-world scenarios Julien et al. (2024); Brenner et al. (2024); Goerigk & Kurtz (2025). Bertsimas & Kim (2024) pioneered an ML-based framework that accelerates the solution process for ARO. Their insight is to view the ARO problem's solution structure as a strategy that could be predicted by a trained ML model. Their method involves training a set of ARO instances using traditional CCG methods and learning the optimal strategies for including here-and-now decisions, worst-case scenarios, and wait-and-see actions. On new ARO instances, the trained model returns a policy orders of magnitude faster than classical iterative solvers. An issue with ML methods is that while they can drastically reduce online solution times, they generally provide approximate solutions; rigorous worst-case guarantees can be difficult to establish without additional theoretical structures. Building on ML-based approaches, Dumouchelle et al. (2023) neuralizes CCG by replacing the worst-case uncertainty selection with a learned surrogate of second-stage objective. The surrogate is embedded as a piecewise-linear model in the MP, compatible with MILP solvers Fischetti & Jo (2017); Serra et al. (2018), yielding substantial speedups. Achieving reliable performance, however, requires careful neural architectural choices which are readily embeddable as an MILP.

Embedding neural networks (NNs) into MILPs is impractical, as it requires introducing binary activation variables for each neuron and additional linking constraints (e.g., big-M) at every layer. Thus the MILP model size increases at least linearly with the neuron count and, in practice, the branch-and-bound search explodes as the network deepens Fischetti & Jo (2017); Serra et al. (2018). Big-M constraints are easy to formulate but require tight pre-activation bounds; loose bounds yield weak relaxations and large search trees. For large neural architectures like transformers, the resulting MILP can blow up in memory and time, undermining the speedups the surrogate was meant to provide Tong et al. (2024).

## 1.2 CONTRIBUTIONS

In this paper, we introduce a novel penalized two-phase decomposition strategy for addressing the Master Problem (MP) within the CCG framework for ARO. Specifically, the algorithm circumvents the need to embed the neural networks as mixed-integer linear constraints by splitting the MP into a penalized relaxed phase-1 candidate selection for an uncertainty and a phase-2 NN-based worst-case candidate verification. Building on this, the primary contributions of our work are as follows:

**Decoupled NN Architecture:** Proposing separation of NN inference from the core optimization problem, enabling the use of sophisticated NNs (including those with advanced activation functions) without introducing MILP encoding overhead. This architectural decoupling ensures scalability and faster solutions for high-dimensional ARO problems.

**Stabilized Phase-1 selection** The MP's phase-1 biases the uncertainty selection towards possible worst-case scenarios. This is done through a severity score obtained from penalty calculated for each scenario based on the problem instance. The penalized relaxation stabilizes candidate selection, accelerating convergence by steering the CCG loop toward near worst-case realizations.

**Certifiable Lower Bounds:** Establishes strict lower objective bounds through scenario generation, overcoming a key limitation of existing ML-based ARO approaches that lack formal guarantees.

**Performance Gains:** The proposed framework achieves $10$–$10^2\times$ faster convergence than state-of-the-art methods on average across real and synthetic robust optimization problems. The overall algorithm terminates in finitely many iterations (at most the number of candidate scenarios).

The paper is organized as follows: Section 2 introduces key preliminaries. Section 3 presents the two-phase decomposition framework for MP, including its ML architecture and integration with CCG. Section 4 covers the NN training process. Section 5 evaluates the framework on two-stage robust knapsack and power grid unit commitment problems, comparing runtime and solution quality against benchmarks. Finally, Section 6 presents key insights and future directions.

## 2 PRELIMINARIES

In this study, we adopt a general robust optimization framework where the uncertainties are defined as discrete sets $\widehat{\Xi}$. We reformulate 1 into an extended ARO form as shown in Lefebvre et al. (2023):

$$\min_{\boldsymbol{x} \in X,\, \theta,\, \boldsymbol{y_\xi}} \quad \theta \tag{2a}$$

$$\text{s.t.} \quad \boldsymbol{c}^\top \boldsymbol{x} + \boldsymbol{d}^\top \boldsymbol{y_\xi} \leq \theta, \qquad \forall \boldsymbol{\xi} \in \widehat{\Xi} \tag{2b}$$

$$\boldsymbol{T}(\boldsymbol{\xi})\boldsymbol{x} + \boldsymbol{W}\boldsymbol{y_\xi} \leq h(\boldsymbol{\xi}), \qquad \forall \boldsymbol{\xi} \in \widehat{\Xi} \tag{2c}$$

$$\boldsymbol{y_\xi} \in Y(\boldsymbol{x}, \boldsymbol{\xi}), \qquad \forall \boldsymbol{\xi} \in \widehat{\Xi}. \tag{2d}$$

The extended ARO formulation reduces the $\min - \max - \min$ problem into a single-level $\min$ problem by enumerating uncertainties in $\widehat{\Xi}$. However, this enumeration can introduce a large number of variables $\boldsymbol{y_\xi}$ and constraints (2b)–(2c), increasing computational complexity. In the next section, we discuss algorithms designed to mitigate this challenge.

### 2.1 COLUMN-AND-CONSTRAINT GENERATION (CCG)

The CCG algorithm is a prominent iterative method for tackling medium-scale two-stage ARO problems Zhao & Zeng (2012). It addresses the computational complexity arising from enumeration in (2) by iteratively constructing constraints and variables within a master-adversarial framework. Rather than considering all scenarios, the CCG algorithm focuses on the most critical scenarios in the MP at each iteration, $t \in \mathbb{N}$, using a smaller discrete restricted uncertainty set $\Xi_t \subseteq \widehat{\Xi}$. The MP at iteration $t$ computes the solution $(\boldsymbol{x}_t, \theta_t)$ and is formulated as follows:

$$\mathcal{P} := \min_{\boldsymbol{x} \in X,\, \theta,\, \boldsymbol{y_\xi}} \quad \theta \tag{3a}$$

$$\text{s.t.} \quad \boldsymbol{c}^\top \boldsymbol{x} + \boldsymbol{d}^\top \boldsymbol{y_\xi} \leqslant \theta \qquad \forall \boldsymbol{\xi} \in \Xi_t \tag{3b}$$

$$\boldsymbol{T}(\boldsymbol{\xi})\boldsymbol{x} + \boldsymbol{W}\boldsymbol{y_\xi} \leqslant h(\boldsymbol{\xi}) \qquad\qquad \forall \boldsymbol{\xi} \in \Xi_t \qquad\qquad (3c)$$

$$\boldsymbol{y_\xi} \in Y(\boldsymbol{x}, \boldsymbol{\xi}) \qquad\qquad \forall \boldsymbol{\xi} \in \Xi_t \qquad\qquad (3d)$$

Until a convergence criterion is met, injection of newer uncertainties in the MP takes place by solving the AP which takes in as input the fixed MP solution $\boldsymbol{x}_t$, defined as,

$$Q(\boldsymbol{x}_t) := \max_{\boldsymbol{\xi} \in \hat{\Xi}} \; \min_{\boldsymbol{y_\xi} \in Y(\boldsymbol{x}, \boldsymbol{\xi})} \left\{ \boldsymbol{c}^\top \boldsymbol{x} + \boldsymbol{d}^\top \boldsymbol{y_\xi} : \boldsymbol{W}\boldsymbol{y_\xi} \leqslant h(\boldsymbol{\xi}) - \boldsymbol{T}(\boldsymbol{\xi})\boldsymbol{x^t} \right\}. \qquad (4)$$

The solution to this problem provides $\boldsymbol{\xi}^t$ and the value of the AP as $Q_t$ (in Appendix B Algorithm 3).

## 2.2 Computational challenges of CCG

The iterative structure of CCG, while theoretically sound, suffers from significant computational bottlenecks in large-scale problems or those with integer recourse variables Zhao & Zeng (2012). Even with polyhedral uncertainty and convex second-stage decisions, the AP must be solved as an MILP in each iteration due to bilinear complementarity constraints Zeng & Zhao (2013) or heuristically for bilinear objectives Bertsimas et al. (2012), resulting in intractable representations.

In the MP, every iteration appends a new constraint and variable (for each $\boldsymbol{\xi} \in \Xi_t$) leading to longer solve times especially for integer decisions. Even if individual MP/AP solves are tractable, the total runtime scales with the number of iterations and is thus prohibitive for large problems.

## 3 ML Approximation of CCG

For fixed decisions $\hat{\boldsymbol{x}}$ and uncertainty $\hat{\xi}$, the innermost $\min$ is the following value-function problem,

$$V(\widehat{\boldsymbol{x}}, \widehat{\boldsymbol{\xi}}) := \min_{\boldsymbol{y}} \{ \boldsymbol{c}^\top \widehat{\boldsymbol{x}} + \boldsymbol{d}^\top \boldsymbol{y} : \boldsymbol{W}\boldsymbol{y} \leqslant h(\widehat{\boldsymbol{\xi}}) - \boldsymbol{T}(\widehat{\boldsymbol{\xi}})\widehat{\boldsymbol{x}} \}, \quad \boldsymbol{\xi}^t \; \leftarrow \; \arg\max_{\boldsymbol{\xi}_i \in \hat{\Xi}} V(\boldsymbol{x}^t, \boldsymbol{\xi}_i).$$

During the $t^{\text{th}}$ CCG iteration, if $V(\boldsymbol{x}_t, \boldsymbol{\xi}_i)$ is available for all $\boldsymbol{\xi}_i \in \hat{\Xi}$ either exactly or via function approximation, the adversarial problem in Equation (4) simplifies to enumerating $\hat{\Xi}$ to identify the worst-case uncertainty. The selected $\boldsymbol{\xi}^t$ is then appended to the MP set $\Xi_t$ for the next CCG iteration.

In this work, we approximate the AP's $V(\hat{\boldsymbol{x}}, \hat{\boldsymbol{\xi}})$, value of the inner $\min$ problem by NN model similar to Dumouchelle et al. (2023). While being an approximate solution, this method avoids bi-linearity and integer-valued variables in AP, replacing it with forward passes through the NN with weights $\Theta$. By training the NN on the representative set of here-and-now decisions and uncertainty $(\widehat{\boldsymbol{x}}, \widehat{\boldsymbol{\xi}})$ from previously recorded data $\mathcal{D}$, the NN model learns a mapping from input to the value of $V(\widehat{\boldsymbol{x}}, \hat{\boldsymbol{\xi}})$.

$$\boldsymbol{\xi}^t \leftarrow \arg\max_{\boldsymbol{\xi} \in \hat{\Xi}} V_\Theta(\boldsymbol{x^t}, \boldsymbol{\xi}) \qquad\qquad (5)$$

As new uncertainties are introduced at each iteration $t$, expanding the uncertainty set $\Xi_t$, the MP correspondingly increases in size. This growth poses computational challenges, particularly when integer variables or a large number of second-stage decision variables are involved, leading to potential solution intractability. To overcome these issues, we propose a two-phase *selection* and *verification* scheme that decomposes the MP and solves it iteratively.

### 3.1 MP: Two-Phase Decomposition

The MP is broken into two key components: a phase-1 penalized Relaxed Master Problem (RMP) that selects a candidate worst-case scenario driven by severity score and a verification step that determines if it is truly the most damaging. If not, a no-good cutting plane is added to refine the MP's feasible region. This selection-verification cycle repeats until the MP solution aligns with the most likely worst-case scenario. Below, we first detail *Candidate Selection*, followed by *Candidate Verification*.

### 3.1.1 MP Phase-1: Selection Phase (severity-weighted)

In the selection phase, the MP is relaxed by omitting the complicating constraints (3b)-(3c). Phase-1 selects a single *active* scenario from the current candidate set $\bar{\Xi}_t$ and a corresponding pair $(\boldsymbol{x}, \boldsymbol{y})$ that

is feasible for that scenario, while gently steering the choice toward *severe* scenarios. We solve the following one–hot selection model (the active uncertainty equals exactly one candidate):

$$\mathcal{P}_1 := \min_{\boldsymbol{x},\,\boldsymbol{y},\,\boldsymbol{\xi}_a,\,\boldsymbol{z}} \quad \boldsymbol{c}^\top \boldsymbol{x} + \boldsymbol{d}^\top \boldsymbol{y} \;-\; \lambda \cdot \sum_{i=1}^{|\Xi_t|} S_\phi(\boldsymbol{\xi}_i)\, z_i, \tag{6a}$$

$$\text{s.t.} \quad \boldsymbol{T}(\boldsymbol{\xi}_a)\,\boldsymbol{x} + \boldsymbol{W}\,\boldsymbol{y} \;\leqslant\; \boldsymbol{h}(\boldsymbol{\xi}_a), \tag{6b}$$

$$\boldsymbol{\xi}_a \;=\; \sum_{i=1}^{|\Xi_t|} z_i\,\boldsymbol{\xi}_i, \qquad \sum_{i=1}^{|\Xi_t|} z_i \;=\; 1, \tag{6c}$$

$$\boldsymbol{\xi}_a^\ell \;\leqslant\; \boldsymbol{\xi}_a \;\leqslant\; \boldsymbol{\xi}_a^u, \tag{6d}$$

$$z_i \in \{0, 1\}, \quad i = 1, \ldots, |\Xi_t|. \tag{6e}$$

Here, $z$ is a one–hot vector that selects the active candidate, $\boldsymbol{\xi}_a$ denotes the selected uncertainty, and $S_\phi(\boldsymbol{\xi}) \in [0, 1]$ is a fixed severity score that depends on $xi$ and the problem instance parameters. We learn a separate regressor $S_\phi(\boldsymbol{\xi}) \in [0, 1]$ with weights $\phi$ on the same training corpus as the second-stage NN $V_\Theta$, but with the decision $x$ *marginalized*. This makes $S_\phi$ instance–uncertainty dependent while being policy-agnostic, since it aggregates $Q(x, \boldsymbol{\xi})$ over a design pool of first-stage decisions. The severity penalty $-\lambda \cdot \sum_i S_\phi(\boldsymbol{\xi}_i) z_i$ biases the objective towards worst-case scenarios. Also, instead of having constraint blocks for each $\boldsymbol{\xi}$, only one constraint equation 6b for $\boldsymbol{\xi}_a$ is used reducing the problem size.

We set $\lambda = \lambda_{\text{mult}} \cdot R$ where $\lambda_{\text{mult}} \geq 0$ is a user-defined multiplier and $R$ is a data-driven reference scale (e.g., a robust inter-quantile spread $R = q_{95\%} - q_{5\%}$ of the raw severity targets or predictions). The full recipe of targets, features, and calibration details of the severity score and $R$ appears in Appendix E. In experiments we sweep $\lambda_{\text{mult}}$ on a grid to find the best value.

**Proposition 1** (Lower Bound). *Let $(\boldsymbol{x}^*, \boldsymbol{z}^*)$ solve $\mathcal{P}_1$ in (6) with $\lambda > 0$ to optimality, and $i^*$ satisfy $z_{i^*}^* = 1$. If $\mathrm{Opt}(\mathcal{P})$ is an optimal value of $\mathcal{P}$ in (3), then:*

$$\mathrm{Opt}\big(\mathcal{P}_1 \mid \lambda = 0, \boldsymbol{z} = \boldsymbol{z}^*\big) = \min_{\boldsymbol{x},\boldsymbol{y}} \big\{ \boldsymbol{c}^\top \boldsymbol{x} + \boldsymbol{d}^\top \boldsymbol{y} \,\big|\, \boldsymbol{T}(\boldsymbol{\xi}_{i^*})\boldsymbol{x} + \boldsymbol{W}\boldsymbol{y} \leq \boldsymbol{h}(\boldsymbol{\xi}_{i^*}) \big\} \leqslant \mathrm{Opt}(\mathcal{P}). \tag{7}$$

*Proof sketch.* See Appendix A for a full proof.

Crucially, Proposition 1 certifies the lower bound only after setting $\lambda = 0$ (removing the scenario-weighting penalty); accordingly, $\mathcal{P}_1$ serves solely to select a worst-case scenario $i^*$, not to certify the bound itself.

### 3.1.2 MP Phase-2: Verification Phase

A candidate worst-case uncertainty for MP, denoted by $\xi_a^*$, is identified from $\mathcal{P}_1$ along with $\boldsymbol{x}^*$. Phase-1 may select a severe but non-worst scenario in $\Xi_t$. We therefore enforce a verification step to confirm its validity. The verification is done using the following:

$$\mathcal{P}_2(\boldsymbol{x}^*) := \max_{\boldsymbol{\xi} \in \Xi_t} \min_{\boldsymbol{y} \in Y(\boldsymbol{x}^*, \xi)} \Big\{ \boldsymbol{c}^\top \boldsymbol{x}^* + \boldsymbol{d}^\top \boldsymbol{y} \;\; \boldsymbol{W}\boldsymbol{y} \leqslant h(\boldsymbol{\xi}) - \boldsymbol{T}(\boldsymbol{\xi})\boldsymbol{x}^* \Big\},$$

where $\boldsymbol{x}^*$ is the Phase-1 decision. This verification problem searches for an uncertainty $\widehat{\boldsymbol{\xi}} \in \Xi_t$ that maximizes the worst-case cost associated with $\boldsymbol{x}^*$. If $\hat{\boldsymbol{\xi}} = \boldsymbol{\xi}_a^*$, then the candidate uncertainty is confirmed as the worst-case scenario for the current iteration. Otherwise, if $\boldsymbol{\xi} \neq \boldsymbol{\xi}_a^*$, a new constraint (e.g., $z_i = 0$ corresponding to the rejected scenario) is added to $\mathcal{P}_1$ and it is resolved, effectively acting as a *cutting-plane* that excludes the suboptimal candidate and tightens the LB.

**Phase-2 Approximation**: Note $\mathcal{P}_2(\boldsymbol{x}^*)$ solves a similar problem as compared to $Q(\boldsymbol{x}^t)$ albeit on a smaller set of MP uncertainties for a fixed $\boldsymbol{x}^*$ from selection phase $\mathcal{P}_1$. Thus, the NN model used in the second-stage of CCG is also applied as an approximation in the MP phase-2 denoted as $\mathcal{P}_2^{\text{approx}}$.

### 3.2 ML-accelerated CCG

The proposed ML-Accelerated CCG framework integrates NN approximations into both the MP and AP replacing costly NN-embedded mixed-integer solves. Algorithm 2 outlines the full procedure, which iteratively refines lower and upper bound (UB) until convergence.

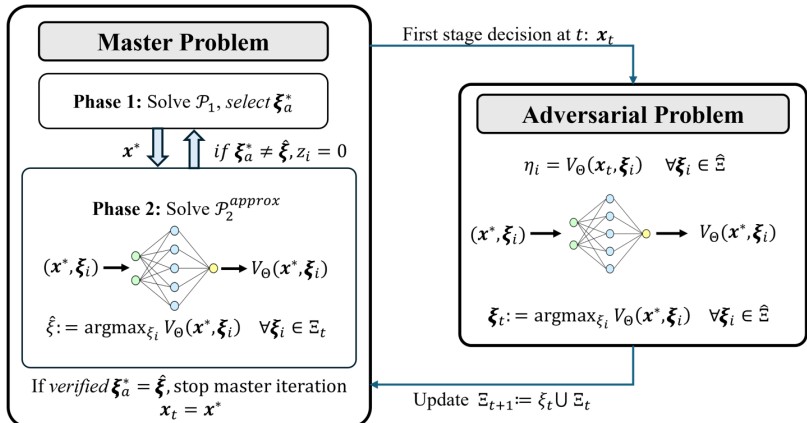

Figure 1: Overview of the proposed Machine Learning-Accelerated CCG algorithm.

**Convergence.** After phase-1 selection and phase-2 verification, we fix the selected scenario $\boldsymbol{\xi}_t^{\text{in}}$ and solve the corresponding single–scenario problem *exactly* (with the unpenalized objective) to obtain the MP's certified LB. We then evaluate the global worst-case for $x_t$ over $\widehat{\Xi}$, yielding the certified UB. Since the severity penalty only biases Phase–1 selection and never enters these certificates, the gap $\text{UB} - \text{LB}$ decreases monotonically and, with a finite scenario pool, the algorithm terminates once $\text{UB} - \text{LB} \leq \varepsilon$, where $\varepsilon$ is the tolerance.

To verify the approximation quality of the phase-2 and AP in the ML-Accelerated CCG by an NN, we consider "Accelerated-CCG" where both phase-2 and AP are solved exactly using the optimal solution to the value-function problem. Due to the exact phase-2 and AP solves, Accelerated CCG serves as the benchmark for comparing against the ML-accelerated variant. This framework, showed in Algorithm 5 is critical for enabling and benchmarking ML approximations while addressing the computational challenges of traditional CCG.

## 4 NEURAL NETWORK TRAINING

### 4.1 DATA GENERATION FOR TRAINING

To train a neural network surrogate for the second-stage cost function, we first generate a representative set of problem instances and uncertainty realizations. Specifically:

**Instance Parameters:** We generate $k$ problem instances $I_i, \forall i \in (1, \ldots, k)$ by sampling parameters, such as cost vectors $\boldsymbol{c}, \boldsymbol{d}$ and constraint matrices $\boldsymbol{T}, \boldsymbol{W}$. As explained in Appendix C and D, sampling distributions are chosen to reflect realistic instances for training.

**Uncertainty Set via Norm Ball:** Let $\bar{\boldsymbol{\xi}}$ represent the nominal (forecasted) uncertainty, with each sampled $\boldsymbol{\xi}$ drawn from a norm ball centered at $\bar{\boldsymbol{\xi}}$, $\boldsymbol{\xi} \in \{\bar{\boldsymbol{\xi}} + \delta : \|\delta\| \leqslant \rho\}$, where $\|\cdot\|$ is a chosen norm (e.g., $\ell_2$ or $\ell_\infty$), and $\rho$ specifies the uncertainty radius. To discretize the uncertainty set:

$$\widehat{\Xi} = \{\bar{\boldsymbol{\xi}} + \delta_i : \|\delta_i\| \leqslant \rho, i = 1, \ldots, n\},$$

where $\delta_i$ is sampled uniformly over the ball.

### 4.2 LABEL COMPUTATION VIA EXACT SOLVES

Each input from the generated data $(\boldsymbol{\xi}, I)$ is used to solve an exact optimization problem:

$$Q(\boldsymbol{x}_t, \boldsymbol{\xi}) := \min_{\boldsymbol{x}, \boldsymbol{y}} \Big\{ \boldsymbol{c}^\top \boldsymbol{x} + \boldsymbol{d}^\top \boldsymbol{y}, \quad \boldsymbol{T}(\boldsymbol{\xi})\boldsymbol{x} + \boldsymbol{W}\boldsymbol{y} \leqslant \boldsymbol{h}(\boldsymbol{\xi}) \Big\}, \tag{8}$$

using a solver. In the above formulation, first-stage decisions $\boldsymbol{x}$ is a variable. The solution to (8) provides the fixed objective value $V^*$ and $\hat{\boldsymbol{x}}$ for $\boldsymbol{\xi}$ and $I$. This is an expensive task since the NN training requires large number of input data but the solves can be done in an offline distributed manner. Each training instance is stored as $\big((\hat{\boldsymbol{x}}, \boldsymbol{\xi}, I), V^*\big)$ in the dataset $\mathcal{D}$ with $k \times N$ total rows.

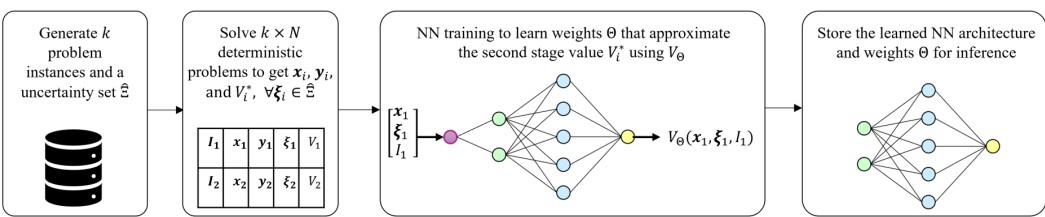

Figure 2: Data generation and training for NN: Training data is generated offline by solving a deterministic program over a tabular combination of instances and uncertainties. The NN model is then trained on the dataset $\mathcal{D}$ for inference in the ML-Accelerated CCG.

### 4.3 NN Training and Inference

A feed-forward NN $V_\Theta(\hat{\boldsymbol{x}}, \boldsymbol{\xi}, I)$, is used to approximate $V^*$. The input layer concatenates $\hat{\boldsymbol{x}}$, $\boldsymbol{\xi}$, and $I$ in one long vector and the output layer predicts a single cost value $V^*$.

$$\mathcal{L}(\Theta) = \frac{1}{|\mathcal{D}|} \sum_{(\hat{\boldsymbol{x}}, \boldsymbol{\xi}) \in \mathcal{D}} \left( V^* - V_\Theta(\hat{\boldsymbol{x}}, \hat{\boldsymbol{\xi}}, I) \right)^2. \tag{9}$$

More advanced architectures, like convolutional or attention-based models, can be used for structured data (see Sec. D.2). After training, the NN model $V_\Theta(\boldsymbol{x}, \boldsymbol{\xi}, I)$ is used for inference in the ML-accelerated CCG algorithm. Since inference is performed with fixed instance parameters, we simplify the notation to $V_\Theta(\boldsymbol{x}, \boldsymbol{\xi})$.

## 5 Computational Results

We evaluate the performance of the proposed ML-accelerated CCG framework against the traditional CCG algorithm, focusing on runtime reduction and solution quality. All optimization and instance testing were performed on a personal laptop with a 2.90 GHz Intel® Core™ i7 CPU and 16 GB of memory. Mixed-Integer Linear Programming problems were solved using Gurobi 11.0, with implementations in Julia via the JuMP package Lubin et al. (2023). NN training and evaluation were conducted on an NVIDIA A30 GPU with 50 GB of memory, using PyTorch 2.4 in Python 3.10.

Next, define "Optimality Gap" quantifies the relative error between the proposed and baseline methods:

$$\text{Optimality Gap} = \frac{O_{\text{proposed}} - O_{\text{baseline}}}{O_{\text{baseline}}} \times 100.$$

Here, $O_{\text{proposed}}$ is the objective value from the proposed accelerated/ML-accelerated CCG framework, while $O_{\text{baseline}}$ is the exact classical CCG solution with respect to the $\hat{\Xi}$ set. A smaller gap indicates a closer alignment with the true robust optimal solution. We also empirically note that the proposed method never violates the LB property from the exact solves, which would otherwise result in over-conservative results.

### 5.1 Case Study: Two-Stage Robust Knapsack

We adopt and test the two-stage robust knapsack problem instances with objective uncertainty from Arslan & Detienne (2022) for Uncorrelated (UN), Weakly Correlated (WC), Almost Strongly Correlated (ASC), and Strongly Correlated (SC) knapsack sizes of 20 to 80 items. The discrete uncertainty set for inference for all the knapsack sizes is taken from Dumouchelle et al. (2023). Since the proposed ML-Accelerated CCG introduces the $\lambda_{mult}$ as a tunable hyperparameter in the penalty term, we perform a grid search to select the value that yields the best trade-off between optimality gap and runtime. For the robust knapsack case study, the NN second-stage models used is presented in Appendix Table 3, the best performance was achieved with $\lambda_{\text{mult}} = 3000.0$ (Appendix E.3), which we fix in all reported experiments.

We evaluate the full two-stage robust knapsack problem, comparing ML-Accelerated CCG and Accelerated CCG against the exact baseline. Figure 3 shows optimality gaps for the robust knapsack,

with a few key observations:

(i) Accuracy: The median optimality gap for each knapsack size remains within $7\%$ for $I = 20, 30$ and within $3\%$ for the sizes greater than 30, while optimality gaps greater than $10\%$ range are rare and reflect trade-offs from NN approximation. Table 2 shows solutions with $0 - 2\%$ optimality gap showing that most of the solutions are tight and closer to the exact objective values.

(ii) LB guarantee: We empirically check that all gaps are non-negative, confirming the proposed method preserves the LB, suggesting the algorithm never produces overly conservative results,

(iii) Reference Performance: Solving phase-2 exactly, rather than using NN-approximated value functions, results in near-zero gaps, confirming the tightness of our two-phase decomposition and the benefit in using larger and better NNs, and

(iv) Fast Solves: The ML-Accelerated CCG achieves orders-of-magnitude faster solve times across all instance sizes, reducing runtimes compared to classical CCG's exact baseline, as shown in Table 1.

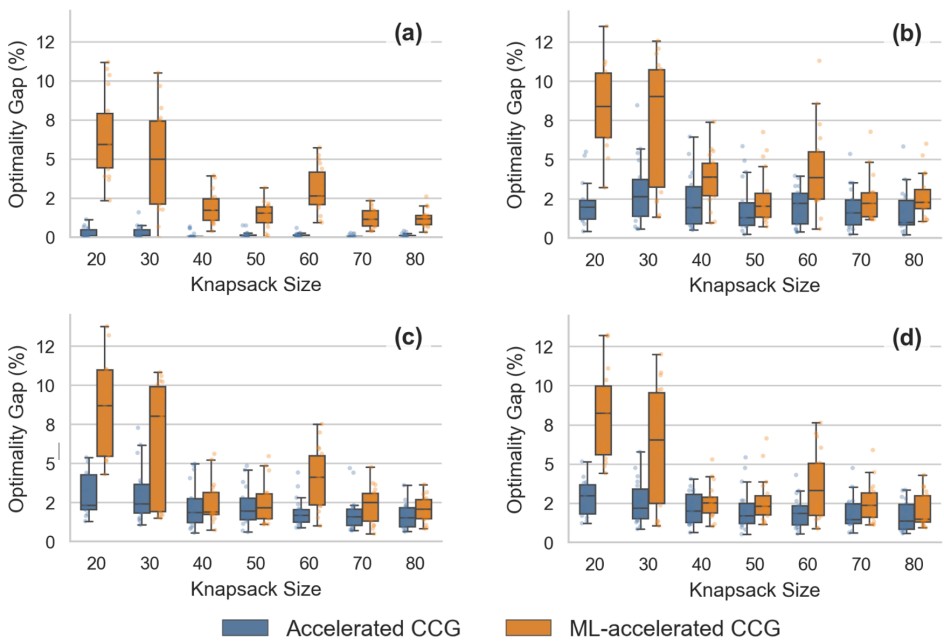

Figure 3: Optimal Gap distribution of the proposed ML-accelerated CCG (orange) and Accelerated CCG (blue) compared to exact CCG solves across problem scales (knapsack item counts $N = 20$ to $80$ and correlation types **(a)** UN, **(b)** WC, **(c)** ASC, **(d)** SC

Table 1: Mean Execution Times (seconds) by Category and Knapsack Size ($I$).

| | UN | | | WC | | | ASC | | | SC | | |
|---|---|---|---|---|---|---|---|---|---|---|---|---|
| $I$ | **Exact** | **ACC.** | **ML.** | **Exact** | **ACC.** | **ML.** | **Exact** | **ACC.** | **ML.** | **Exact** | **ACC.** | **ML.** |
| 20 | 575.8 | 222.6 | 0.14 | 1632.9 | 174.9 | 0.18 | 887.9 | 118.5 | 0.17 | 1311.2 | 166.9 | 0.22 |
| 30 | 545.0 | 237.9 | 0.20 | 2397.2 | 218.3 | 0.25 | 2046.3 | 154.7 | 0.22 | 2737.4 | 212.3 | 0.22 |
| 40 | 843.3 | 324.3 | 0.31 | 2788.6 | 265.6 | 0.36 | 2914.2 | 180.2 | 0.38 | 3319.7 | 206.6 | 0.39 |
| 50 | 575.8 | 307.5 | 0.35 | 3417.6 | 240.3 | 0.43 | 2987.8 | 241.9 | 0.37 | 1909.7 | 199.0 | 0.35 |
| 60 | 1055.4 | 316.5 | 0.44 | 3327.0 | 172.9 | 0.53 | 3413.7 | 120.2 | 0.45 | 2640.4 | 140.5 | 0.44 |
| 70 | 951.2 | 381.3 | 0.54 | 2932.0 | 283.9 | 0.60 | 3561.2 | 272.5 | 0.42 | 3553.7 | 253.3 | 0.51 |
| 80 | 982.0 | 442.6 | 0.54 | 1962.8 | 363.6 | 0.58 | 2651.0 | 319.8 | 0.48 | 2670.9 | 288.7 | 0.51 |

*Exact: Exact baseline,    ACC.: Accelerated CCG,    ML.: ML-accelerated CCG*

## 5.2    CASE STUDY: TWO-STAGE ROBUST UNIT COMMITMENT

We consider the two-stage robust unit commitment (UC) problem with a linearized second-stage and polyhedral demand uncertainty, using the IEEE 6-bus Wu et al. (2009) and 24-bus Ordoudis et al.

(2016) systems as defined in Lorca & Sun (2014); Bertsimas et al. (2012). As a critical problem in power grid operations, UC determines generator schedules while accounting for uncertainty to ensure system reliability and cost efficiency. Solving the CCG algorithm over 24 hours for each generator introduces significant combinatorial complexity, particularly in the 24-bus system with 12 generators, resulting in $3 \times 12 \times 24 = 864$ binary variables and $12 \times 24 \times |\Xi_t|$ continuous variables.

Since decisions for buses (generators and demands) influence one another, the NN model must capture complex, long-range interdependencies within the power grid network. To address this, we propose leveraging a Graph Attention Network (GAT) Veličković et al. (2017) with global attention, where each bus is treated as a node and edges represent transmission line susceptances. The detailed formulation and GAT architecture are presented in Appendix D.

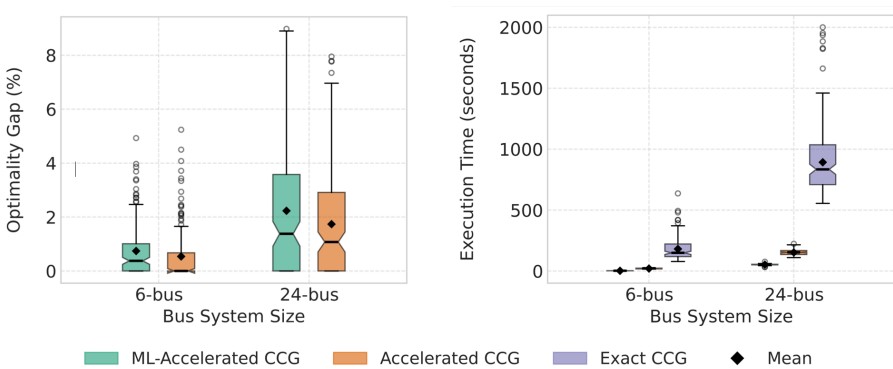

Figure 4: Comparison of solution quality and runtime across methods. The figure shows performance of the proposed ML-Accelerated CCG, the Accelerated CCG against classical CCG baseline. Both solution accuracy and computational time are reported, highlighting that the ML-Accelerated CCG achieves near-identical solution quality while substantially reducing runtime.

In Figure 4, results from 150 instances show that both proposed models achieve optimality gaps within 5% for the 6-bus system and 8.5% for the 24-bus system, with a median gap of 2%. Despite the increased dimensionality of the UC formulation, the method provides significant computational savings over CCG, as shown in Table 6. For the 6-bus system, ML-Accelerated CCG achieves up to a 93× speed-up, while for the larger 24-bus system, it still delivers a 16× speed-up, compared to the traditional CCG. Notably, these gains come with a median optimality gap near 2%, and large number of instances with 0% optimality gap as showed in Table 5 demonstrating that even at larger scales, ML-Accelerated CCG maintains tight LBs (due to relaxations) alongside substantial computational efficiency. As with the knapsack study, we tune the $\lambda_{\text{mult}}$ via grid search to calibrate the bias in phase-1 selection. The best values were found to be $\lambda_{\text{mult}} = 5000$ for the 6-bus UC system and $\lambda_{\text{mult}} = 7500$ for the 24-bus UC system and we will use these values throughout the results.

## 6 CONCLUSIONS

We introduced *LARO*, a learning-accelerated two-phase decomposition framework for two-stage Adaptive Robust Optimization (ARO). By employing a neural-network-based value function approximation, our novel ML-accelerated CCG approach decouples the network from direct solver embeddings, enabling large-scale, mixed-integer ARO while providing finite convergence and stronger LBs than state-of-the-art methods. Numerical experiments on robust knapsack and power grid unit commitment problems reveal substantial speedups—up to $10^3 \times$ for knapsack instances and 16× for a 24-bus system—compared to the baseline CCG, with median optimality gaps under 3%. Overall, our proposed method successfully *balances* the competing demands of *robustness* and *computational scalability*, offering an efficient and flexible alternative for two-stage ARO problems. Future work will explore different types of uncertainty sets, approximate value-function problems with mixed-integer non-convex feasible regions, and develop advanced neural architectures for very large-scale applications.

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

APPENDIX

## A    PROOF OF PROPOSITION 1

*Proof sketch.* Let $\mathcal{P}$ and $\mathcal{P}_1$ be the MP and the relaxed MP, respectively, as defined in equation 3 and equation 6.

**MP $\mathcal{P}$.**

$$\mathcal{P}: \quad \min_{\boldsymbol{x} \in X, \, \theta, \, \{\boldsymbol{y}_{\boldsymbol{\xi}}\}_{\boldsymbol{\xi} \in \Xi_t}} \theta$$
$$\text{s.t.} \quad \boldsymbol{c}^\top \boldsymbol{x} + \boldsymbol{d}^\top \boldsymbol{y}_{\boldsymbol{\xi}} \le \theta, \ \forall \boldsymbol{\xi} \in \Xi_t,$$
$$\boldsymbol{T}(\boldsymbol{\xi})\boldsymbol{x} + \boldsymbol{W}\boldsymbol{y}_{\boldsymbol{\xi}} \le h(\boldsymbol{\xi}), \ \forall \boldsymbol{\xi} \in \Xi_t,$$
$$\boldsymbol{y}_{\boldsymbol{\xi}} \in Y(\boldsymbol{x}, \boldsymbol{\xi}), \ \forall \boldsymbol{\xi} \in \Xi_t.$$

**Relaxed MP $\mathcal{P}_1$.:** The lower bound is evaluated without the penalty term effectively at $\lambda = 0$ after the phase-1 and 2 iterations are complete for a fixed $z^\star$, thus the penalty term disappears from the Relaxed MP objective during LB calculation.

$$\mathcal{P}_1: \quad \min_{\boldsymbol{x}, \, \boldsymbol{y}, \, \boldsymbol{\xi}_a, \, \boldsymbol{z}} \boldsymbol{c}^\top \boldsymbol{x} + \boldsymbol{d}^\top \boldsymbol{y}$$
$$\text{s.t.} \quad \boldsymbol{T}(\boldsymbol{\xi})\boldsymbol{x} + \boldsymbol{W}\boldsymbol{y} \le h(\boldsymbol{\xi}_a),$$
$$\boldsymbol{\xi}_a = \sum_{i=1}^{|\Xi_t|} z_i \boldsymbol{\xi}_i, \quad \sum_{i=1}^{|\Xi_t|} z_i = 1,$$
$$\boldsymbol{\xi}_a^\ell \le \boldsymbol{\xi}_a \le \boldsymbol{\xi}_a^u, \quad z_i \in \{0, 1\}.$$

**Step 1 (feasible sets).** $\mathcal{P}$ enforces constraints for all $\boldsymbol{\xi} \in \Xi_t$; $\mathcal{P}_1$ enforces them for one selected $\boldsymbol{\xi}_a$. Hence $\text{Feas}(\mathcal{P}) \subseteq \text{Feas}(\mathcal{P}_1)$.

**Step 2 (construct a feasible point for $\mathcal{P}_1$).** Let $(\boldsymbol{x}^*, \theta^*, \{\boldsymbol{y}_{\boldsymbol{\xi}}^*\})$ be optimal for $\mathcal{P}$. Pick $\bar{\boldsymbol{\xi}} \in \Xi_t$ attaining $\max_{\boldsymbol{\xi}}\{\boldsymbol{c}^\top \boldsymbol{x}^* + \boldsymbol{d}^\top \boldsymbol{y}_{\boldsymbol{\xi}}^*\}$ and set $\boldsymbol{x} = \boldsymbol{x}^*$, $\boldsymbol{y} = \boldsymbol{y}_{\bar{\boldsymbol{\xi}}}^*$, $\boldsymbol{\xi}_a = \bar{\boldsymbol{\xi}}$, and $z_i = \mathbb{1}\{\boldsymbol{\xi}_i = \bar{\boldsymbol{\xi}}\}$. Then $(\boldsymbol{x}, \boldsymbol{y}, \boldsymbol{\xi}_a, \boldsymbol{z})$ is feasible for $\mathcal{P}_1$.

**Step 3 (objectives).**
$$\boldsymbol{c}^\top \boldsymbol{x} + \boldsymbol{d}^\top \boldsymbol{y} = \boldsymbol{c}^\top \boldsymbol{x}^* + \boldsymbol{d}^\top \boldsymbol{y}_{\bar{\boldsymbol{\xi}}}^* \le \theta^* = \text{Opt}(\mathcal{P}),$$

so minimizing over $\text{Feas}(\mathcal{P}_1)$ gives $\text{Opt}(\mathcal{P}_1 \mid \lambda = 0, \boldsymbol{z} = \boldsymbol{z}^*) \le \text{Opt}(\mathcal{P})$. $\qquad \square$

## B    ALGORITHM

The traditional CCG algorithm described in Section 2.1 is presented as in Algorithm 3:

---

**Algorithm 1** MP with Two Phase Decomposition

---

**Require:** Candidate scenario set $\Xi_t = \{\boldsymbol{\xi}_i\}_{i=1}^{N_t}$;
        NN adversary $V_\Theta(x, \boldsymbol{\xi})$;
        severity scores $S(\boldsymbol{\xi}) \in [0, 1]$;
        penalty scale $R$;
        multiplier $\lambda_{\mathrm{mult}} \geq 0$;
        weight $\lambda \leftarrow \lambda_{\mathrm{mult}} \cdot R$.
1: **Phase 1: Candidate Selection**
2: Solve $\widetilde{\mathcal{P}}_1$ in equation 6 to obtain candidate $(\boldsymbol{x}^*,\ \boldsymbol{\xi}_a^*,\ \theta^*)$
   **Phase 2 (Verification)**
3: **repeat**
4:    Evaluate $\eta_i \leftarrow V_\Theta(\bar{x}, \boldsymbol{\xi}_i)$ for all $\boldsymbol{\xi}_i \in \Xi_t$ and set $j^\star \in \arg\max_i \eta_i, \widehat{\boldsymbol{\xi}} \leftarrow \boldsymbol{\xi}_{j^\star}$.
5:    **if** $\boldsymbol{\xi}_a^* = \widehat{\boldsymbol{\xi}}$ **then**
6:       **break**                ▷ Selection consistent with restricted worst-case
7:    **else**
8:       **if** $|\Xi_t| > 1$ **then**
9:          Add no-good cut forbidding the previous pick: set $z_k = 0$ for $k$ with $\bar{z}_k = 1$.
10:          Re-solve Phase 1.
11:       **else**
12:          Skip the cut to retain feasibility of $\sum_i z_i = 1$.
13:          **break**
14:       **end if**
15:    **end if**
16: **until** consistent
   **Exact LB certification (unpenalized)**       ▷ Severity penalty never appears in certificates
17: Solve the exact single-scenario problem at the verified $\widehat{\boldsymbol{\xi}}$:

$$\mathrm{LB}_t\ =\ \min_{x,y}\ \boldsymbol{c}^\top x + \boldsymbol{d}^\top y \quad \text{s.t.} \quad \boldsymbol{T}(\widehat{\boldsymbol{\xi}})\,x + \boldsymbol{W}\,y \leq \boldsymbol{h}(\widehat{\boldsymbol{\xi}}),$$

   and let $(x_t, y_t)$ be an optimizer. Set $\boldsymbol{\xi}_t \leftarrow \widehat{\boldsymbol{\xi}}$.
18: **Return** $(x_t, \boldsymbol{\xi}_t, \mathrm{LB}_t)$.

---

**Algorithm 2** ML-Accelerated CCG

---

**Require:** Data $(\boldsymbol{c}, \boldsymbol{d}, \boldsymbol{T}(\boldsymbol{\xi}), \boldsymbol{W}, \boldsymbol{h}(\boldsymbol{\xi}))$, full scenario set $\widehat{\widehat{\Xi}}$, NN adversary $V_\Theta(x, \boldsymbol{\xi})$, tolerance $\varepsilon > 0$,
   penalty scale $R$, multiplier $\lambda_{\mathrm{mult}} \geq 0$, severity score $S_\phi$
1: Initialize $t \leftarrow 0$, choose warm-start subset $\Xi_0 \subseteq \widehat{\widehat{\Xi}}$, set UB $\leftarrow +\infty$, LB $\leftarrow -\infty$
2: Set $\lambda \leftarrow \lambda_{\mathrm{mult}} \cdot R$               ▷ weight for Phase–1 bias; not used in certificates
3: **while** UB − LB $> \varepsilon$ **do**
4:    **Step 1: Master Problem**
5:       Call Algorithm 1
6:       Receive $(x_t, \boldsymbol{\xi}_t^{\mathrm{in}}, \mathrm{LB}_t)$       ▷ $\mathrm{LB}_t$ from exact, *unpenalized* single-scenario solve
7:       LB $\leftarrow \max\{$LB, $\mathrm{LB}_t\}$
8:    **Step 2: Adversarial Problem**
9:       Compute $\eta(\boldsymbol{\xi}) = V_\Theta(x_t, \boldsymbol{\xi})$ values $\forall\, \widehat{\widehat{\Xi}}$
10:       $\boldsymbol{\xi}_t^\star \leftarrow \arg\max_{\boldsymbol{\xi} \in \widehat{\Xi}} \eta(\boldsymbol{\xi})$;
11:       Compute $Q(x_t, \boldsymbol{\xi}_t^\star)$ via exact solve;    UB$_t \leftarrow Q(x_t, \boldsymbol{\xi}_t^\star)$
12:       UB $\leftarrow \min\{$UB, UB$_t\}$
13:    **Step 3: Scenario set update**
14:       **if** $\boldsymbol{\xi}_t^\star \notin \Xi_t$ **then** $\Xi_{t+1} \leftarrow \Xi_t \cup \{\boldsymbol{\xi}_t^\star\}$ **else** $\Xi_{t+1} \leftarrow \Xi_t$
15:       $t \leftarrow t + 1$
16: **end while**
17: **Finalize:** $\bar{\theta} \leftarrow$ LB;    $\bar{x} \leftarrow x_t$
18: **return** $(\bar{\theta}, \bar{x})$

---

**Algorithm 3** CCG algorithm

**Require:** $(\boldsymbol{c}, \boldsymbol{d}, \boldsymbol{T}(\boldsymbol{\xi}), \boldsymbol{W}, h(\boldsymbol{\xi}))$ where $\boldsymbol{\xi} \in \widehat{\widehat{\Xi}}$
1: Initialize $t \leftarrow 0, \Xi^t \subset \widehat{\widehat{\Xi}}, UB \leftarrow \infty, LB \leftarrow -\infty$
2: **while** $LB \leq UB$ **do**
3:     Solve MP equation 3, obtain $(x^t, \theta^t)$
4:     $LB \leftarrow \theta^t$
5:     Solve AP equation 4, obtain $(\boldsymbol{\xi}^t, Q^t)$
6:     $UB \leftarrow \min(UB, Q^t); \quad \Xi^t \leftarrow \{\boldsymbol{\xi}^t\} \cup \Xi^t$
7:     $t \leftarrow t + 1$
8: **end while**
9: **return** $\boldsymbol{x}_t, \theta_t$

---

**Algorithm 4** MP: Two-Phase Decomposition (Exact Phase-2)

**Require:** candidate value set $V_\Theta$, master set $\Xi_t$, iteration index $t$
    **Phase 1: Candidate Selection**
1: Solve $\mathcal{P}_1$ to select a candidate $(\boldsymbol{x}^*, \boldsymbol{\xi}_a^*)$
    **Phase 2: Candidate Verification (exact)**
2: Solve $Q(\boldsymbol{x}^*)$ over $\Xi_t$ to obtain $\widehat{\boldsymbol{\xi}}$
3: **if** $\boldsymbol{\xi}_a^* = \widehat{\boldsymbol{\xi}}$ **then**
4:     **return** $(\boldsymbol{x}^*, \boldsymbol{\xi}_a^*, Q(\boldsymbol{x}^*))$
5: **else**
6:     Add cut to $\mathcal{P}_1$: set $z_{k^*} \leftarrow 0$ where $k^* = \{ k \mid z_k = 1 \}$
7:     **restart** Phase 1
8: **end if**

9: **Finalize:** $\theta^* \leftarrow Q(\boldsymbol{x}^*)$
10: **return** $(\boldsymbol{x}^*, \boldsymbol{\xi}_a^*, \theta^*)$

---

**Algorithm 5** Accelerated CCG

**Require:** instance $(\boldsymbol{c}, \boldsymbol{d}, \boldsymbol{T}(\boldsymbol{\xi}), \boldsymbol{W}, h(\boldsymbol{\xi}))$, uncertainty set $\widehat{\widehat{\Xi}}$
**Ensure:** optimal solution $(\bar{\theta}, \bar{\boldsymbol{x}})$
1: **initialize:** $t \leftarrow 0, \Xi_0 \leftarrow \varnothing, UB \leftarrow +\infty, LB \leftarrow -\infty$
2: **while** $LB < UB$ **do**
3:     **Step 1 (MP via Two-Phase):** apply Algorithm 4 with exact Phase-2
4:       obtain $(\boldsymbol{x}^*, \boldsymbol{\xi}_a^*, \theta^*)$ and set $LB \leftarrow \theta^*$

5:     **Step 2 (AP, exact):** solve $Q(\boldsymbol{x}^*)$ over $\widehat{\widehat{\Xi}}$, obtaining worst-case $\boldsymbol{\xi}^{(t)}$ and value $Q^{(t)}$
6:     $UB \leftarrow \min(UB, Q^{(t)})$

7:     **Step 3 (master set update):** $\Xi_{t+1} \leftarrow \Xi_t \cup \{\boldsymbol{\xi}^{(t)}\}; \quad t \leftarrow t + 1$
8: **end while**
9: **finalize:** $\bar{\theta} \leftarrow LB, \quad \bar{\boldsymbol{x}} \leftarrow \boldsymbol{x}^*$
10: **return** $(\bar{\theta}, \bar{\boldsymbol{x}})$

## C  TWO-STAGE ROBUST KNAPSACK PROBLEM

### C.1  MATHEMATICAL FORMULATION

A two-stage robust knapsack problem is considered with a set of $N$ items from which some items $i \in N$ are selected for production. The profit of the items has an uncertain degradation, due to which second-stage decisions of producing as is, repairing, or outsourcing the items have to be made. The complete formulation is:

$$Z := \min_{\boldsymbol{x} \in \{0,1\}^N} \max_{\boldsymbol{\xi} \in \Xi} \min_{\substack{\boldsymbol{y} \in \{0,1\}^N, \\ \boldsymbol{r} \in \{0,1\}^N}} \sum_{i=1}^{N} \Big[ (f_i - \overline{p}_i) x_i + (\hat{p}_i\,\boldsymbol{\xi}_i - f_i) y_i - \hat{p}_i\,\boldsymbol{\xi}_i\, r_i \Big] \tag{10a}$$

$$\text{s.t.} \quad \sum_{i=1}^{N} \big( c_i\, y_i + t_i\, r_i \big) \leqslant C, \tag{10b}$$

$$r_i \leqslant y_i \leqslant x_i, \quad \forall\, i = 1, \dots, N, \tag{10c}$$

where $\Xi = \Big\{ \boldsymbol{\xi} \in [0,1]^N : \sum_{i=1}^{N} \boldsymbol{\xi}_i \leqslant \Gamma \Big\}$ describes the uncertainty set. In equation 10a, the first-stage decision $\boldsymbol{x}$ selects items to produce. The inner (minimization) problem determines the optimal second-stage responses after uncertainty $\boldsymbol{\xi}$ is realized: producing an item as is, $(y_i = 1)$ generates a profit that depends on the degradation $(\overline{p}_i - \boldsymbol{\xi}_i \hat{p}_i)$, whereas repairing it $(r_i = 1)$ recovers the full profit at the expense of extra resource $t_i$. Or the item can be outsourced for a profit of $(\overline{p}_i - f_i)$.

The capacity constraint equation 10b limits the overall resource usage, and the logical constraint equation 10c guarantees that an item is only kept if produced, and can only be repaired if it is kept. The formulation captures the adversarial nature of the problem by considering the worst-case degradation over the uncertainty set $\Xi$.

### C.2  INSTANCE GENERATION

The problem instances for a specific instance size $N$ of uncorrelated knapsack are generated through the Algorithm 6. All uniform distributions are denoted by $\mathcal{U}(a,b)$, representing values drawn uniformly from the interval $[a,b]$. In Algorithm 6, we begin by specifying an instance size $N$ and initializing global parameters: the cost UB $R$, the base capacity $H$, a capacity scaling factor $h$ chosen from $\{40, 80\}$, a degradation factor $\delta$ from $\{0.1, 0.5, 1\}$, and a knapsack budget $\Gamma$ from $\{0.1 \times N, 0.2 \times N, 0.3 \times N\}$. For each item $i$, we generate its cost $c_i$ from a uniform distribution on $[1, R]$, then compute the total adjusted capacity $C$ as $h \times H + \sum_{i=1}^{N} c_i$. We sample the nominal price $\overline{p}_i$ from $[1, R]$, the adjusted price $\hat{p}_i$ from the interval $\left[ \frac{\overline{p}_i(1-\delta)}{2}, \frac{\overline{p}_i(1+\delta)}{2} \right]$, the fixed cost $f_i$ from $[1.1\,\overline{p}_i, 1.5\,\overline{p}_i]$, and the processing time $t_i$ from $[1, c_i]$. The resulting instance is $\big( \boldsymbol{c}, \overline{\boldsymbol{p}}, \hat{\boldsymbol{p}}, \boldsymbol{f}, \boldsymbol{t}, C \big)$, which serves as input for learning the value function of the robust knapsack problem.

### C.3  UNCERTAINTY GENERATION FOR THE KNAPSACK PROBLEM

In Algorithm 7, a random generator is seeded to ensure reproducibility, and each scenario vector is sampled from the Dirichlet distribution with unit parameters, reflecting a base probability distribution that sums to one. The vector is then scaled by the budget $\Gamma$ to produce a nonnegative vector whose components sum to $\Gamma$. Repeating this process for $M$ scenarios yields a diverse set of uncertainty vectors, each representing a valid realization under the knapsack's budget constraint.

We generate a set $\mathcal{I}$ of size $|\mathcal{I}| = 250$ and, for each instance in $\mathcal{I}$, a set of 50 uncertainties, yielding a total of 7,500 unique uncertainty realizations. We then vary $\Gamma$ across 11 discrete values to construct feasible first-stage decisions. The resulting dataset, denoted by $\mathcal{D}$, thus has cardinality $|\mathcal{D}| = 250 \times 50 \times 11 = 137,500$ for each instance size.

**Algorithm 6** Knapsack Instance Generation Algorithm (Training)

---

**Require:** Instance size $N$
1: **Initialize instance parameters**
2: $R \leftarrow 1000$                                            ▷ cost UB
3: $H \leftarrow 100$                                     ▷ capacity parameter
4: $h \leftarrow$ uniformly at random from $\{40, 80\}$
5: $\delta \leftarrow$ uniformly at random from $\{0.1, 0.5, 1\}$
6: $\Gamma \leftarrow$ uniformly at random from $\{0.1N, 0.2N, 0.3N\}$
7: **for** each item $i \in \{1, \ldots, N\}$ **do**
8:     $c_i \sim \mathcal{U}(1, R)$
9: **end for**
10: $C \leftarrow h \cdot H + \sum_{i=1}^{N} c_i$
11: **for** each item $i \in \{1, \ldots, N\}$ **do**
12:     $\overline{p}_i \sim \mathcal{U}(1, R)$
13:     $\hat{p}_i \sim \mathcal{U}\left(\frac{\overline{p}_i(1-\delta)}{2}, \frac{\overline{p}_i(1+\delta)}{2}\right)$
14:     $f_i \sim \mathcal{U}(1.1\,\overline{p}_i, 1.5\,\overline{p}_i)$
15:     $t_i \sim \mathcal{U}(1, c_i)$
16: **end for**
17: **return** $I := (\boldsymbol{c}, \overline{\boldsymbol{p}}, \hat{\boldsymbol{p}}, \boldsymbol{f}, \boldsymbol{t}, C)$

---

**Algorithm 7** Uncertainty Generation via Dirichlet Distribution (Training)

---

**Require:** number of scenarios $M$, uncertainty budget $\Gamma$, instance size $N$, random seed
**Ensure:** set $\{\boldsymbol{\xi}^{(1)}, \ldots, \boldsymbol{\xi}^{(M)}\}$ with $\sum_{i=1}^{N} \boldsymbol{\xi}_i^{(k)} = \Gamma$ and $\boldsymbol{\xi}_i^{(k)} \geq 0$
1: initialize RNG with *seed*
2: **for** $k \leftarrow 1$ to $M$ **do**
3:     sample $s^{(k)} \sim \text{Dirichlet}(\alpha)$ where $\alpha = \mathbf{1}_N$
4:     scale: $\boldsymbol{\xi}^{(k)} \leftarrow \Gamma \cdot s^{(k)}$
5:     store $\boldsymbol{\xi}^{(k)}$ in output set
6: **end for**
7: **return** $\{\boldsymbol{\xi}^{(k)}\}_{k=1}^{M}$

---

Table 2: Percentage of near-optimal solutions $\leq 2\%$ optimality gap) by Category and Knapsack Size out of 18 instances per cell

| I | UN | | WC | | ASC | | SC | |
|---|---|---|---|---|---|---|---|---|
| | ACC. | ML | ACC. | ML | ACC. | ML | ACC. | ML |
| 20 | 18 (100.0%) | 6 (33.3%) | 14 (77.8%) | 1 (5.6%) | 12 (66.7%) | 1 (5.6%) | 14 (77.8%) | 2 (11.1%) |
| 30 | 18 (100.0%) | 9 (50.0%) | 15 (83.3%) | 6 (33.3%) | 15 (83.3%) | 6 (33.3%) | 17 (94.4%) | 7 (38.9%) |
| 40 | 18 (100.0%) | 18 (100.0%) | 16 (88.9%) | 14 (77.8%) | 18 (100.0%) | 16 (88.9%) | 18 (100.0%) | 17 (94.4%) |
| 50 | 18 (100.0%) | 18 (100.0%) | 17 (94.4%) | 15 (83.3%) | 18 (100.0%) | 17 (94.4%) | 17 (94.4%) | 16 (88.9%) |
| 60 | 18 (100.0%) | 16 (88.9%) | 18 (100.0%) | 12 (66.7%) | 18 (100.0%) | 11 (61.1%) | 18 (100.0%) | 12 (66.7%) |
| 70 | 18 (100.0%) | 18 (100.0%) | 17 (94.4%) | 17 (94.4%) | 18 (100.0%) | 18 (100.0%) | 18 (100.0%) | 17 (94.4%) |
| 80 | 18 (100.0%) | 18 (100.0%) | 17 (94.4%) | 16 (88.9%) | 18 (100.0%) | 18 (100.0%) | 18 (100.0%) | 18 (100.0%) |

*ACC.: Accelerated CCG, ML: ML-accelerated CCG*

| Instance size | NN Structure | ReLU Neurons | Training MAPE (%) | Epochs |
|---|---|---|---|---|
| 20 | 1 linear + 4 ReLU | 140, 110, 32, 8 | 88.2 | 300 |
| 30 | 1 linear + 5 ReLU | 155, 128, 64, 16, 8 | 91.6 | 300 |
| 40 | 1 linear + 5 ReLU | 340, 128, 110, 32, 8 | 87.5 | 300 |
| 50 | 1 linear + 5 ReLU | 500, 256, 128, 32, 8 | 90.3 | 400 |
| 60 | 1 linear + 5 ReLU | 410, 256, 128, 64, 12 | 89.1 | 400 |
| 70 | 1 linear + 6 ReLU | 540, 256, 256, 128, 64, 16 | 92.0 | 400 |
| 80 | 1 linear + 6 ReLU | 610, 400, 256, 128, 64, 16 | 90.7 | 500 |

Table 3: Neural network hyperparameters and training performance (MAPE) for knapsack instance sizes.

**Algorithm 8** Data Generation for NN Training of the Knapsack Problem

---

**Require:** number of instances $|\mathcal{I}|$, instance size $N$, uncertainties per variant $M$

 **Initialize:** empty database $\mathcal{D}$

 **Base Instance Generation**

1: **for** $k \leftarrow 1$ to $|\mathcal{I}|$ **do**

2:      generate base instance $\mathcal{I}_k$ using Algorithm 6

3:      store parameters $(\boldsymbol{c}^k, \overline{\boldsymbol{p}}^k, \widehat{\boldsymbol{p}}^k, \boldsymbol{f}^k, \boldsymbol{t}^k, C^k, \Gamma_k)$

4: **end for**

 **Instance Variation**

5: **for** each $\mathcal{I}_k \in \{\mathcal{I}_1, \ldots, \mathcal{I}_{|\mathcal{I}|}\}$ **do**

6:      generate 11 budget variants $\{\mathcal{I}_k^1, \ldots, \mathcal{I}_k^{11}\}$ with

          $\Gamma_k^m \leftarrow \big(0.75 + 0.025(m-1)\big)\,\Gamma_k \quad (m = 1, \ldots, 11)$

7:      generate $M$ uncertainty vectors $\{\boldsymbol{\xi}_k^1, \ldots, \boldsymbol{\xi}_k^M\}$ using Algorithm 7

8: **end for**

 **Solution Computation**

9: **for** each variant $\mathcal{I}_k^m$ **do**

10:      **for** each $\boldsymbol{\xi}_k^i \in \{\boldsymbol{\xi}_k^1, \ldots, \boldsymbol{\xi}_k^M\}$ **do**

11:          solve Formulation 10 with $(\Gamma_k^m, \boldsymbol{\xi}_k^i)$

12:          record $\big(\boldsymbol{x}_k^{mi}, \boldsymbol{y}_k^{mi}, \boldsymbol{r}_k^{mi}, Z_k^{mi}, t_k^{mi}\big)$

13:          $\mathcal{D} \leftarrow \mathcal{D} \cup \big\{(\boldsymbol{x}_k^{mi}, \boldsymbol{y}_k^{mi}, \boldsymbol{r}_k^{mi}, Z_k^{mi}, t_k^{mi})\big\}$

14:      **end for**

15: **end for**

16: **return** $\mathcal{D}$

---

## D  TWO-STAGE ROBUST UC

Consider a $T$-period network-constrained unit commitment (UC) problem with a set of $M$ buses, and a set of $N$ generators distributed among these buses. For each bus $m \in \{1, \ldots, M\}$, let $\mathbb{N}_m$ denote the set of generators connected to bus $m$. The time horizon is discretized into $T$ time periods, indexed by $t \in \{1, \ldots, T\}$.

Each generator $i \in \mathbb{N}_m$ has an associated set of parameters:

- $S_i^m$ and $W_i^m$: start-up and shut-down costs,
- $G_i^m$ and $H_i^m$: minimum up-time and minimum down-time requirements,
- $L_i^m$ and $U_i^m$: minimum and maximum power output when switched on,
- $V_i^m$ and $B_i^m$: ramp-up and ramp-down rate limits,
- $\overline{V}_i^m$ and $\overline{B}_i^m$: start-up and shut-down ramp rate limits.

We denote by $y_{i,t}^m$ a binary variable that is equal to 1 if generator $i$ at bus $m$ is on during period $t$ and 0 otherwise. We use $u_{i,t}^m$ and $v_{i,t}^m$ to indicate start-up and shut-down events, respectively, of generator $i$ at bus $m$ in period $t$. The variable $x_{i,t}^m$ represents the power output of generator $i$ at bus $m$ and time $t$.

Let $\mathbb{A}$ be the set of transmission lines, where each line $(i, j) \in \mathbb{A}$ connects two buses $i$ and $j$ and has a capacity $C_{i,j}$. We model power flows using a DC power flow approximation. Accordingly, we introduce voltage angle variables $\omega_{m,t}$ at each bus $m$ and time $t$, and let $B_{i,j}$ denote the susceptance of line $(i, j)$. The power flow on the line $(i, j)$ at time $t$ is denoted by $f_{(i,j),t}$.

Demand at each bus $m$ and time $t$ is given by $\boldsymbol{\xi}_t^m$, which is subject to uncertainty. Let $\widehat{\Xi}$ denote the uncertainty set of possible demand realizations $\{\boldsymbol{\xi}_t^m\}$. We then formulate a two-stage robust optimization problem in which the first-stage chooses the unit commitment decisions $\{y_{i,t}^m, u_{i,t}^m, v_{i,t}^m\}$, and the second-stage, after observing the demand realization in $\widehat{\Xi}$, determines the dispatch decisions $\{x_{i,t}^m, f_{(i,j),t}, \omega_{m,t}\}$ to minimize operating costs while satisfying all network constraints.

The resulting two-stage robust UC model is formulated as follows.

$$\min_{\boldsymbol{y},\boldsymbol{u},\boldsymbol{v}} \quad \sum_{t=1}^{T} \sum_{m=1}^{M} \sum_{i \in \mathbb{N}_m} \left( S_i^m u_{i,t}^m + W_i^m v_{i,t}^m \right) + \max_{\boldsymbol{\xi} \in \widehat{\Xi}} \min_{(\boldsymbol{x},\boldsymbol{\omega},\boldsymbol{\Omega}) \in \mathcal{X}(\boldsymbol{y},\boldsymbol{\xi})} \sum_{t=1}^{T} \sum_{m=1}^{M} \sum_{i \in \mathbb{N}_m} \Omega_{i,t}^m \quad (11a)$$

s.t.

$$- y_{i,t-1}^m + y_{i,t}^m - y_{i,k}^m \leqslant 0, \quad 1 \leqslant k - (t-1) \leqslant G_i^m, \quad \forall m, \forall i \in \mathbb{N}_m, \forall t, \quad (11b)$$

$$y_{i,t-1}^m - y_{i,t}^m + y_{i,k}^m \leqslant 1, \quad 1 \leqslant k - (t-1) \leqslant H_i^m, \quad \forall m, \forall i \in \mathbb{N}_m, \forall t, \quad (11c)$$

$$- y_{i,t-1}^m + y_{i,t}^m - u_{i,t}^m \leqslant 0, \quad \forall m, \forall i \in \mathbb{N}_m, \forall t, \quad (11d)$$

$$y_{i,t-1}^m - y_{i,t}^m - v_{i,t}^m \leqslant 0, \quad \forall m, \forall i \in \mathbb{N}_m, \forall t. \quad (11e)$$

Define the second-stage feasible set $\mathcal{X}(\boldsymbol{y}, \boldsymbol{\xi})$ for a given $(\boldsymbol{y}, \boldsymbol{\xi})$ as:

$$\mathcal{X}(\boldsymbol{y}, \boldsymbol{\xi}) = \Big\{ (\boldsymbol{x}, \boldsymbol{\omega}, \boldsymbol{\Omega}): \quad L_i^m y_{i,t}^m \leqslant x_{i,t}^m \leqslant U_i^m y_{i,t}^m, \quad \forall m, \forall i \in \mathbb{N}_m, \forall t, \quad (11f)$$

$$x_{i,t}^m - x_{i,t-1}^m \leqslant (2 - y_{i,t-1}^m - y_{i,t}^m) \overline{V}_i^m + (1 + y_{i,t-1}^m - y_{i,t}^m) V_i^m, \quad \forall m, i, t, \quad (11g)$$

$$x_{i,t-1}^m - x_{i,t}^m \leqslant (2 - y_{i,t-1}^m - y_{i,t}^m) \overline{B}_i^m + (1 - y_{i,t-1}^m + y_{i,t}^m) B_i^m, \quad \forall m, i, t, \quad (11h)$$

$$f_{(i,j),t} = B_{i,j} (\omega_{i,t} - \omega_{j,t}), \quad \forall (i,j) \in \mathbb{A}, \forall t, \quad (11i)$$

$$- C_{i,j} \leqslant f_{(i,j),t} \leqslant C_{i,j}, \quad \forall (i,j) \in \mathbb{A}, \forall t, \quad (11j)$$

$$\sum_{i \in \mathbb{N}_m} x_{i,t}^m - d_t^m = \sum_{j:(m,j) \in \mathbb{A}} f_{(m,j),t}, \quad \forall m, \forall t, \quad (11k)$$

$$\omega_{\text{slack},t} = 0, \quad \forall t, \Big\}. \tag{11l}$$

## D.1 DATA GENERATION

The NN training for the UC problem takes as input the instance values $I_i :=$ $(S_i^m, W_i^m, G_i^m, H_i^m, L_i^m, U_i^m, V_i^m, B_i^m, \hat{V}_i^m, \hat{B}_i^m)$, uncertainty set $\boldsymbol{\xi}_i$, and the first-stage decisions as bus features and the susceptance matrix $B$. The data generation process is defined as follows.

**Instance and Uncertainty Generation**

In this method, 150 instance data for the Unit Commitment (UC) problem for 6-bus and 24-bus system are generated by applying perturbations within the radius of a norm ball (see Algorithms 9, 10). The radius is selected as a percentage of the total sum of the normalized nominal values. Similarly the we create 1000 and 2500 uncertainties for 6-bus and 24-bus system respectively by using the nominal data is obtained from Ordoudis et al. (2016) and Wu et al. (2009).

---

**Algorithm 9** UC Instance Generation

---

**Require:** nominal vector $\boldsymbol{I}_{\text{nom}} \in \mathbb{R}_{\geq 0}^n$, perturbation factor $p \in [0,1]$, number of instances $|\mathcal{I}|$, bounds $\boldsymbol{I}_{\min}, \boldsymbol{I}_{\max} \in \mathbb{R}_{\geq 0}^n$

1: $\boldsymbol{v} \leftarrow \boldsymbol{I}_{\text{nom}}/\|\boldsymbol{I}_{\text{nom}}\|_1$          ▷ normalize
2: $S \leftarrow \|\boldsymbol{I}_{\text{nom}}\|_1$          ▷ total scale
3: $\Delta \leftarrow p \cdot S$          ▷ L1 perturbation budget
4: $\mathcal{I}_{\text{sim}} \leftarrow \varnothing$          ▷ output set
5: **for** $i \leftarrow 1$ to $|\mathcal{I}|$ **do**
6:      sample $\boldsymbol{r} \sim \text{Uniform}(-1,1)^n$
7:      $\boldsymbol{r} \leftarrow \boldsymbol{r}/\|\boldsymbol{r}\|_1$          ▷ direction in L1
8:      $\boldsymbol{r}_{\text{pert}} \leftarrow \Delta \cdot \boldsymbol{r}$          ▷ scale to budget
9:      $\boldsymbol{v}_{\text{pert}} \leftarrow \boldsymbol{v} + \boldsymbol{r}_{\text{pert}}$          ▷ perturb normalized vector
10:     $\boldsymbol{I}^{(i)} \leftarrow S \cdot \boldsymbol{v}_{\text{pert}}$          ▷ denormalize
11:     $\boldsymbol{I}^{(i)} \leftarrow \text{clip}\big(\boldsymbol{I}^{(i)}, \boldsymbol{I}_{\min}, \boldsymbol{I}_{\max}\big)$          ▷ box constraints
12:     $\mathcal{I}_{\text{sim}} \leftarrow \mathcal{I}_{\text{sim}} \cup \{\boldsymbol{I}^{(i)}\}$
13: **end for**
14: **return** $\mathcal{I}_{\text{sim}}$

---

**Algorithm 10** Demand Uncertainty Generation

---

**Require:** nominal demands per bus $\{\boldsymbol{\xi}_b \in \mathbb{R}^T\}_{b \in \mathbb{B}}$, norm radius $\alpha > 0$, number of scenarios $N_{\text{samples}} \in \mathbb{N}$

**Ensure:** scenario set $\widehat{\Xi} = \{\Xi^{(k)}\}_{k=1}^{N_{\text{samples}}}$, where $\Xi^{(k)} = \{\boldsymbol{\xi}_b^{(k)}\}_{b \in \mathbb{B}}$ and $\boldsymbol{\xi}_b^{(k)} \in \mathbb{R}_{\geq 0}^T$

1: $\widehat{\Xi} \leftarrow \varnothing$
2: **for** $k \leftarrow 1$ to $N_{\text{samples}}$ **do**
3:      $\Xi^{(k)} \leftarrow \varnothing$
4:      **for each** $b \in \mathbb{B}$ **do**
5:          $\boldsymbol{v} \leftarrow \boldsymbol{\xi}_b$          ▷ nominal demand (length $T$, e.g., $T{=}24$)
6:          $r \leftarrow \alpha \|\boldsymbol{v}\|_2$          ▷ bus-specific perturbation radius
7:          sample $\boldsymbol{z} \sim \mathcal{N}(\boldsymbol{0}, \boldsymbol{I}_T); \quad \boldsymbol{z} \leftarrow \boldsymbol{z}/\|\boldsymbol{z}\|_2$
8:          sample $\rho \sim \text{Uniform}(0, r)$
9:          $\boldsymbol{\delta} \leftarrow \rho \boldsymbol{z}$
10:         $\boldsymbol{\xi}_b^{(k)} \leftarrow \max(\boldsymbol{0}, \boldsymbol{v} + \boldsymbol{\delta})$          ▷ clip at zero to enforce nonnegativity
11:         $\Xi^{(k)} \leftarrow \Xi^{(k)} \cup \{\boldsymbol{\xi}_b^{(k)}\}$
12:      **end for**
13:      $\widehat{\Xi} \leftarrow \widehat{\Xi} \cup \{\Xi^{(k)}\}$
14: **end for**
15: **return** $\widehat{\Xi}$

---

## D.2 GRAPH ATTENTION ARCHITECTURE

We employ a Graph Attention Network (GAT) model that combines bus features $\boldsymbol{h}_i$ for $i^{th}$ bus with eigenvalue-based Positional Encodings (PEs) illustrated in the Figure 5. The eigenvalue PEs described in Dwivedi et al. (2021); You et al. (2019) are obtained by the eigen decomposition of the susceptance matrix $B$, and the first $k$ eigenvector-based PEs are transformed via a small two-layer MLP to match the input dimensionality of the node features. The node features and transformed PEs are then summed element-wise and passed through three consecutive GAT layers.

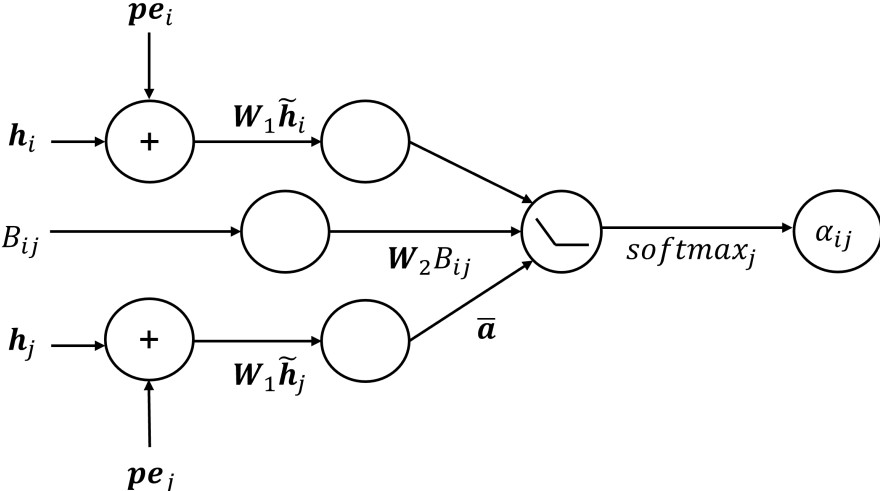

Figure 5: An illustration of the GAT model's attention $\alpha_{i,j}$ between two buses (nodes) $i$ and $j$. The input feature vector $h_i$ is element-wise added to $pe_i$. The softmax score $\alpha_{i,j}$ is calculated by concatenating $W_1\tilde{\boldsymbol{h}}_i$ and $W_1\tilde{\boldsymbol{h}}_j$ along with the susceptance transformation $W_2 B_{i,j}$.

Each GAT layer leverages multi-head attention as described in Veličković et al. (2017) but we concatenate a trainable transformation of the susceptance values between the transfomed bus feature vectors. We update the attention calculations as follows:

$$\alpha_{ij} = \frac{\exp\left(\text{LeakyReLU}\left(\bar{\boldsymbol{a}}^T[\boldsymbol{W_1}\tilde{\boldsymbol{h}}_i|\boldsymbol{W_2}B_{i,m}|\boldsymbol{W_1}\tilde{\boldsymbol{h}}_j]\right)\right)}{\sum_{m\in\{1,\ldots,M\}}\exp\left(\text{LeakyReLU}\left(\bar{\boldsymbol{a}}^T[\boldsymbol{W_1}\tilde{\boldsymbol{h}}_i|\boldsymbol{W_2}B_{i,m}|\boldsymbol{W_1}\tilde{\boldsymbol{h}}_m]\right)\right)}$$

The attention score for the bus $i$ is calculated for every other bus in the grid $m \in \{1, \ldots, M\}$, also denoted in the denominator of the attention softmax score formula. This leads to global attention being calculated instead of local attention as usually done in GCNs Zhang et al. (2019); Wu et al. (2019).

We use global attention to determine the long range dependencies between buses along with residual connections and layer normalization to stabilize training across layers. After the GAT layers, the node embeddings are aggregated (via mean pooling) and fed into a final MLP regressor, which outputs a single scalar approximating the value of the second-stage objective. Table 4 summarizes the main hyperparameters used in our experiments.

The feature vector $h_{\text{in}}$ only depends on the generator property, demand, and first-stage decison made only on that bus, and thus the feature vector size is invariant to the size of the power-grid. Due to the fixed input vector size and mean pooling operation the GAT architecture thus is also invariant to the size of the power grid and can adapt to any bus system. The training and validation error of the 24-bus system for each epoch is presented in the Figure 6

| Hyperparameter | Value |
|---|---|
| $h_{in}$ (Input feature dimension) | 58 |
| $\tilde{h}$ (Hidden dimension) | 128 |
| $h_{out}$ (Output dimension) | 1 |
| num attention heads | 4 |
| MLP (Susceptance) | 4 |
| $\alpha$ (LeakyReLU slope) | 0.1 |
| dropout | 0.15 |
| norm | LayerNorm |
| $k$ (Eigen PE dimension) | 24 |
| mlp_hidden | 128 |
| residual | True |
| concat_heads | True |
| epochs | 2500 |
| batch size | 512 |
| train:test split | 90:10 |

Table 4: Key hyperparameters of the GAT model.

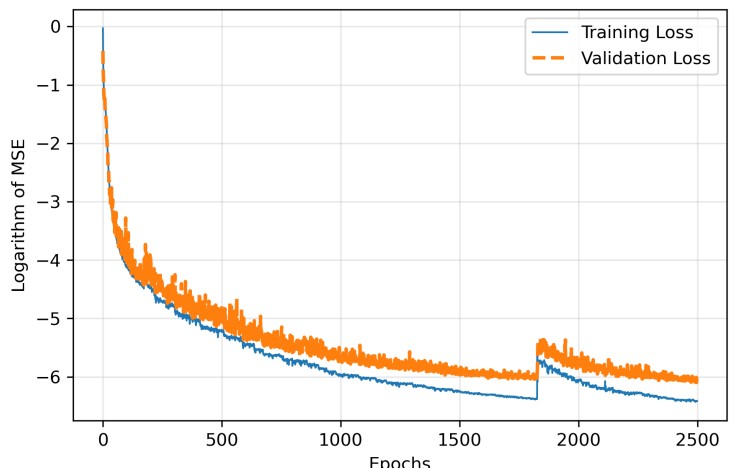

Figure 6: Training vs. validation MSE loss curve (log scale) of GAT NN for the 24-bus system.

| Case | 6-bus | 24-bus |
|---|---|---|
| ML-Accelerated CCG vs. CCG | 64 | 51 |
| Accelerated CCG vs. CCG | 92 | 82 |

Table 5: Number of UC problem instances with exact solutions ($\approx 0\%$ optimality gap) out of 150 total testing instances.

| System size | Method | Min. (s) | Max. (s) | Mean (s) |
|---|---|---|---|---|
| 6-Bus | ML-Accelerated CCG | 1.33 | 2.64 | 1.93 |
| | Accelerated CCG | 15.55 | 25.11 | 20.96 |
| | CCG | 78.67 | 637.39 | 181.09 |
| 24-Bus | ML-Accelerated CCG | 32.82 | 76.77 | 52.80 |
| | Accelerated CCG | 110.15 | 223.90 | 153.19 |
| | CCG | 555.49 | 1999.31 | 892.87 |

Table 6: Computation time (seconds) comparison of ML-accelerated CCG, Accelerated CCG, and CCG for the 6-bus and 24-bus UC problem.

# E  SEVERITY SCORE AND PENALTY SCALING

We describe the recipes used in our experiments to (i) compute an *instance–uncertainty* severity score $S(\boldsymbol{\xi}; I) \in [0, 1]$ that biases Phase-1 selection, and (ii) set the penalty weight $\lambda = \lambda_{\mathrm{mult}} \cdot R$ by a percentile spread. Both are computed **offline** from data and are kept **fixed** during each master solve. Importantly, $S(\boldsymbol{\xi}; I)$ depends only on the instance parameters $I$ and the uncertainty $\boldsymbol{\xi}$—not on decision variables—so adding the penalty in Phase-1 does not alter feasibility or certificates.

**Note:** For brevity in the main text we often drop the instance argument of the problem and write $S(\boldsymbol{\xi})$ and $V_\Theta(x, \boldsymbol{\xi})$; however, all learned quantities are conditioned on the instance, i.e., $S(\boldsymbol{\xi}; I)$ and $V_\Theta(x, \boldsymbol{\xi}; I)$, with $I$ provided as part of the feature vector to $V_\Theta$.

## E.1  SEVERITY SCORE FROM RECOURSE LOSS

For each training instance $I$ and each uncertainty $\boldsymbol{\xi}$, we define a *raw recourse-loss target* by marginalizing the decision over a small, fixed reference pool $\mathcal{X}_{\mathrm{ref}}(I)$:

$$
v(I, \boldsymbol{\xi}) := \frac{1}{|\mathcal{X}_{\mathrm{ref}}(I)|} \sum_{x \in \mathcal{X}_{\mathrm{ref}}(I)} Q(x, \boldsymbol{\xi}; I), \qquad \text{where } Q(\cdot, \cdot; I) \text{ is the exact second-stage value.}
$$
(12)

We then calibrate $v(I, \cdot)$ to $[0, 1]$ *per instance* using train-fold empirical $5^{\mathrm{th}}$ and $95^{\mathrm{th}}$ percentiles over uncertainties:

$$
S(\boldsymbol{\xi}; I) := \mathrm{clip}\left( \frac{v(I, \boldsymbol{\xi}) - q_{5\%}(I)}{\max\{ q_{95\%}(I) - q_{5\%}(I), \epsilon \}}, 0, 1 \right), \quad \epsilon = 10^{-8},
$$
(13)

where $q_\alpha(\boldsymbol{\omega})$ denotes the $\alpha$-quantile of $\{ v(I, \boldsymbol{\xi}) : \xi \in \widehat{\Xi}^{\mathrm{train}} \}$. This yields a monotone, outlier-robust normalization that preserves the ranking of uncertainties for each instance. The resulting $S(\cdot; I)$ is used *only* in the Phase-1 objective bias; all certificates (LB/UB) are computed with the unpenalized objective.

## E.2  PENALTY WEIGHT VIA PERCENTILE SPREAD (SINGLE RECIPE)

We set the weight as $\lambda = \lambda_{\mathrm{mult}} \cdot R$ with a user multiplier $\lambda_{\mathrm{mult}} \geq 0$ and a *single* data-driven scale $R$ taken as a percentile difference over the *entire* training corpus $\mathcal{D}_{\mathrm{train}}$ of instance–uncertainty pairs:

$$
R := q_{95\%}\left( \{ v(I, \boldsymbol{\xi}) : (I, \boldsymbol{\xi}) \in \mathcal{D}_{\mathrm{train}} \} \right) - q_{5\%}\left( \{ v(I, \boldsymbol{\xi}) : (I, \boldsymbol{\xi}) \in \mathcal{D}_{\mathrm{train}} \} \right), \qquad \lambda = \lambda_{\mathrm{mult}} \cdot R.
$$
(14)

Here $v(I, \boldsymbol{\xi})$ is defined by equation 12, computed on the train fold only. This choice makes $\lambda$ scale-free and comparable across instance families and sizes; we sweep $\lambda_{\mathrm{mult}}$ on a small grid in experiments and report the best setting in the main text, with full grids in the appendix.

**Implementation details.** In our experiments, the severity score $S(\boldsymbol{\xi}; I)$ is produced by a *three-layer* feed-forward neural network (MLP with hidden widths $64 \rightarrow 64$, ReLU activations, and a final sigmoid), trained on the recourse-loss targets $v(I, \boldsymbol{\xi})$ from equation 12 with squared loss, early stopping, and weight decay; the network output is then calibrated to $[0, 1]$ via equation 13. For the penalty weight, we use $\lambda = \lambda_{\mathrm{mult}} \cdot R$ with $R$ equal to the *percentile spread* in equation 14; to stabilize this estimate we employ a simple *ensemble quantile estimator* (bootstrap aggregation of the $5^{\mathrm{th}}$ and $95^{\mathrm{th}}$ percentiles, reporting the median spread across resamples). The best value of $\lambda_{mult}$ is obtained using a grid search.

## E.3  GRID SEARCH FOR THE $\lambda$-MULTIPLIER

We tune the Lagrangian bias via a simple grid search over $\lambda_{\mathrm{mult}}$ and report the average optimality gap (lower is better) aggregated across all knapsack instances. As shown in Fig. 7, the curve is shallow around its minimum, indicating robustness to small deviations. We fix $\lambda_{\mathrm{mult}} = 3000$ for the knapsack experiments. For UC, separate sweeps per system yield $\lambda_{\mathrm{mult}} = 5000$ (6-bus) and $\lambda_{\mathrm{mult}} = 7500$ (24-bus).

1188
1189
1190
1191
1192
1193
1194
1195
1196
1197
1198
1199
1200
1201
1202
1203
1204
1205
1206
1207
1208
1209
1210
1211
1212
1213
1214
1215
1216
1217
1218
1219
1220
1221
1222
1223
1224
1225
1226
1227
1228
1229
1230
1231
1232
1233
1234
1235
1236
1237
1238
1239
1240
1241

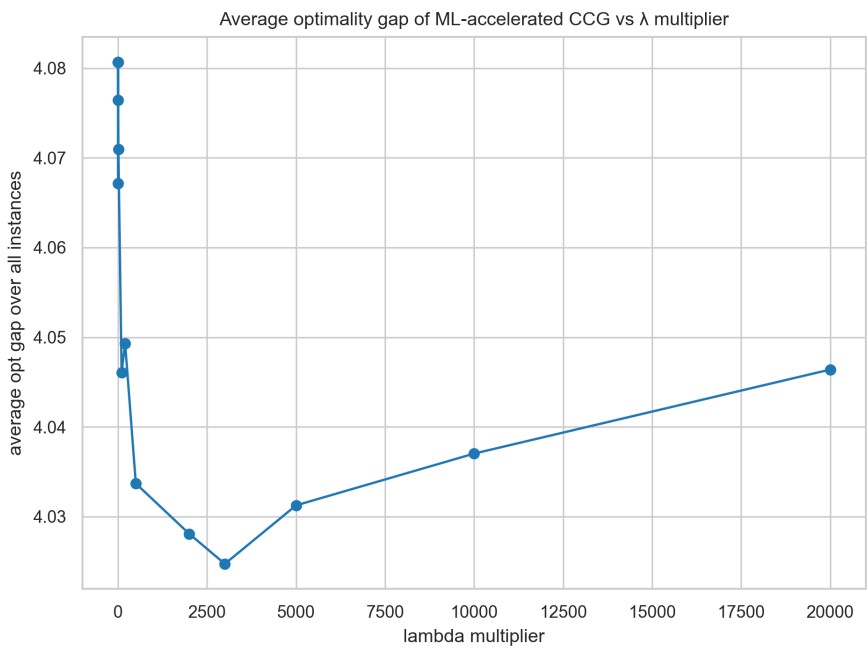

Figure 7: Average optimality gap of ML-Accelerated CCG vs. $\lambda_{\text{mult}}$ across all knapsack instances. The minimum occurs near $\lambda_{\text{mult}}$=3000, which we use in the main results.

