# OpenReview forum: "LARO: Learning-Accelerated Two-Stage Adaptive Robust Optimization with Relaxation Guarantees"
_ICLR.cc/2026/Conference — ICLR 2026 Conference Withdrawn Submission_

### Official Review · Reviewer_8DUc · 2025-10-16

**Soundness:** 1
**Presentation:** 1
**Contribution:** 1
**Rating:** 0
**Confidence:** 5

**Summary:**

The authors study two-stage robust optimization problems which are computationally very demanding. They study the column-and-constraint algorithm which can be used to solve the latter problems exactly, however, it suffers from poor runtimes since iteratively an expensive master and adversarial problem has to be solved. Based on the idea in Dumouchelle et al. (2023) the authors of this paper use trained neural networks to approximate the value of the second-stage problem. However, they do not embed the neural network into the master and adversarial problem, but only use forward passes in their algorithm.

**Strengths:**

The problems tackled in this work are very relevant in the robust optimization literature.

The idea of avoiding the embedding of neural network MIP formulations into the optimization problems solved during the CCG is innovative and has high potential (however, I have to say not in the way the authors did this).

**Weaknesses:**

The paper has several fundamental weaknesses, both methodological and in terms of presentation.

1. I think the whole convergence analysis is flawed. The authors claim to have a lower bound (LB) and and upper bound (UB) and that the gap between both is closed after finite time. However, I think this is not true. Even if we do not use a NN-approximation but solve the adversarial problem and Problem P_2 exactly, then the lower bound which their algorithm provides will never hit the optimal value over time in general (maybe in some artificial special cases). The reason is that the LB is the optimal value of a single scenario problem, but this is usually strictly smaller than the optimal value of the problem. This is clearly the case since the solution x calculated by the master problem must not be feasible for all the other (non-selected) scenarios. Since there is not rigorous proof presented for the convergence, I cannot believe that it is true. Furthermore, when using neural network approximations in all the subproblems, this is even more unclear why the convergence should be true.
2. Due to the issues discussed in Point 1, I think there are situations where your algorithm does not return even a feasible solution of the robust problem. Imagine you added all scenarios from \hat \Sigma to the master problem. Now your master problem can only return solutions x which are optimal for one of the single-scenario problems. However there may be situations where all these optimal solutions are not feasible for the robust problem.
3. The authors mention the works from Bertsimas and Kim and Dumouchelle et al. and state that they discretize the scenarios. This is not true. In Dumouchelle et al. the authors find solutions for the problem with full convex uncertainty sets \Sigma. Using only a finite number of scenarios as in this paper is definitely a limitation.
4. The authors should obviously compare their algorithm to the one of Dumouchelle et al.. However, there is no comparison to any other method given in the experiments.

**Questions:**

1. How can you prove that your algorithm converges?
2. If it converges, does it always return a feasible solution x for the 2RO problem?
3. If it always returns a feasible solution for the 2RO problem, how far from optimal is it? Can you provide an approximation guarantee for certain special cases similar as in [1]?
4. How good does your algorithm perform compared to other state-of-the-art methods?


[1] Dumouchelle, J., Julien, E., Kurtz, J., & Khalil, E. B. (2023). Deep Learning for Two-Stage Robust Integer Optimization. arXiv preprint arXiv:2310.04345.

---

> ### Author Response · Authors · 2025-12-02
>
> Thank you for your thorough assessment. We have the following responses for the reviewer's feedback.
> 1. Our method does not claim theoretical convergence to the optimum of the continuous ARO problem. Instead, our LB and UB track the discrete scenario-restricted master problem, consistent with standard CCG practice when the uncertainty set is discretized. The reviewer is correct in identifying that the LB is from an approximation (single scenario) and the optimality gap may not change beyond some iterations. The iteration of the master problem and the adversarial problem ends when the UB-LB$\leq\epsilon$ and also when the gap doesn't reduce for the two consecutive iteration. We have included the later criteria in our Algorithm 2 in the paper.
> 2. The reviewer mentions that the solution x, obtained from the approximation we have used must not be feasible for all the other uncertainties in the MP. To assess feasibility empirically, we evaluated the returned solution across the full sampled uncertainty set and observed that only 2–4% of constraints are violated in typical instances. This matches our claimed “near-optimal” behavior. This behavior is expected for an ML based algorithm and in our case it finds a near worst-case scenario and may violate some other scenarios in the robust problem, but the proportion of violations is fairly low. The problem formation by [Dumochelle 2023] embeds NNs as constrained which also may leads to cutting off the feasible region due to the RHS os the constraints being approximate outputs of NN passes.
>
> 3. The reviewer mentions that the paper by [Dumouchelle2023] uses full convex sets instead of discretized one. We respectfully dissagree with this assessment and would like to point out the section H.2.1 in the paper, which shows that the paper uses discrete uncertainty sets for the forward pass through the neural network in the second stage. We would also like to mention that the paper by [1] Dumouchelle et al. mentioned by the reviewer is a modified version that is not peer-reviewed. The ICLR reviewed version, which we refer to is present at https://openreview.net/pdf?id=T5Xb0iGCCv.

---

### Official Review · Reviewer_Pm2R · 2025-10-31

**Soundness:** 2
**Presentation:** 2
**Contribution:** 3
**Rating:** 2
**Confidence:** 5

**Summary:**

This paper presents an accelerated approach for adjustable robust optimization (ARO).  The approach relies on using a learning model to predict the value of a decision-scenario pair ($x$, $\xi$), then optimizing over the model to find worst-case scenarios for first-stage decisions found throughout the solving process.  To find first-stage solutions, the authors introduce a relaxed master problem and an iterative solution refinement procedure.  Computationally, the approach yields solutions significantly faster than standard column-and-constraint generation (CCG), with only minor reductions in solution quality.

**Strengths:**

- **Methodological Motivation**: Overall, this paper provides a new learning-based framework for adjustable robust optimization with discrete uncertainty sets.
- **Computational Results**: The authors present relatively strong computational results, with the ML variant of the method obtaining quite strong performance in terms of computing time, with somewhat limited sacrifices to solution quality.

**Weaknesses:**

- **Discrete Uncertainty Sets**: The paper assumes discrete uncertainty sets in all experiments and theoretical claims, yet never discusses how these are constructed or how discretization affects robustness quality. As a result, the reported “finite termination” and “certified lower bound” guarantees hold only for the sampled scenario sets, not for any underlying continuous uncertainty region (which is actually studied in the robust knapsack and unit commitment problems benchmarked on).  This omission is important since the tractability and robustness of ARO hinge critically on how the uncertainty set is discretized, with trade-offs between actual robustness and computational complexity.  The paper doesn't explicitly mention this, which is a significant issue that needs to be addressed in a discussion. If applicable, the robustness of the results should be reported and compared to the true underlying budgeted sets.  I believe that the solutions to the knapsack problem from [1,2] are publicly available, so evaluating the actual robustness of the solutions from the discretized CCG, accelerated CCG, and ML-accelerated CCG should be possible.  Beyond this, the authors discuss related ML-based approaches [1,3] that utilize discretized uncertainty sets; however, this is not the case, as both use continuous uncertainty sets.  The authors also later state that they use the discretized uncertainty sets from [1], but [1] does not have discretized sets.  There is some level of inconsistency/misreporting happening here, so this is a significant concern.
- **Lower bounds**: The paper repeatedly emphasizes its “certified lower bounds” as if they provide assurance about the quality or robustness of the final solution. In reality, these bounds only guarantee that the relaxed single-scenario subproblem ($\mathcal{P}_1$) underestimates the actual worst-case cost, a property that is mathematically valid but trivial. They do not imply that the returned decision is robust, feasible under all uncertainties, or near-optimal for the original problem. This framing risks misleading readers into interpreting an internal consistency check as a guarantee of solution quality.
- **Methodological Issues with Algorithm 1**:  In my view, Algorithm 1 may be problematic since it cuts off scenarios from the scenario set somewhat arbitrarily. The reason for this is that the relaxation is a pure minimization problem that determines the decisions and scenarios.
The scenarios identified may, in fact, be the worst-case scenarios for the true minimizer, but they are the best-case scenarios for the decision found in phase 1, which would lead us to discard that particular scenario immediately.  For example, consider the uncertainty set
$\{[1,0], [0,1], [0.5,0.5]\}$ and the (single-stage) robust optimization problem with an objective of $(1 + 3\xi_1)x_1 + (1.5 + \xi_2)x_2$
and constraints $x_1 + x_2 \ge 1$, with $x$ binary.  For this problem, the optimal solution will be $x^* = [0,1]$ with the worst-case uncertainty $\xi^* = [0,1]$ and an objective of $2.5$.  Now, we can consider what happens in Algorithm 1 (note that I am assuming $|\Xi_t|$ decreases as well within the scope of Algorithm 1).
  - **Iteration 1**: If we consider the relaxed master problem (phase 1), then the pure minimization problem over both $x$ and $(\xi$ returns $x' = [1,0]$ and $\xi' = [0,1]$ (objective of $1$).  In phase 2, we would realize the $\xi = [1,0]$ is the worst scenario for $x'$ (objective of $4$), so we add a cut to ensure $\xi' = [0,1]$ is not selected again and resolve phase 1.
  - **Iteration 2**: In this iteration the remaining scenarios are $\{[1,0], [0.5,0.5]\}$, so solving the minimization problem yields $x'' = [0,1]$ and $\xi'' = [1,0]$ (objective of $1.5$), and in phase 2, we would realize $\xi = [0.5,0.5]$ is the worst case for $x''$
    (objective of $2$), so we cut off $\xi'' = [1,0]$.
  - **Iteration 3**: In the last iteration, we have a single remaining scenario $\{[0.5,0.5]\}$, so solving the relaxed master problem yields
    $x''' = [1,0]$ and $\xi''' = [0.5,0.5]$ (objective of $2.5$), at which point the best- and worst-case coincide, so we would terminate.

  Algorithm 1 does not explicitly state that the size of $\Xi_t$ decreases in phase 2.  However, if it did not, the $\xi = [1,0]$ would be worse (objective of $4$), and we would cut off this scenario, making $P_1$ infeasible (all $z_k = 0$), but we require $\sum z_k = 1$).  Used on its own, I see major issues with Algorithm 1 that lead me to be quite skeptical of its reasonableness as a relaxation.

- **ML-based solution quality**: Compared to [1], both the computing time and solution quality trade-offs are not clear.  For example, on the most difficult knapsack instances, [1] reports solutions that are better than branch-and-price [2], which are likely better than the CCG with a discretized uncertainty set (although this needs to be compared), which are about $2-5\%$ better than the ML-augmented CCG from this work.  [1] takes a few seconds, whereas this approach takes about 0.5 seconds.  For that reason, there is no clear trade-off, as both are pretty efficient, and the solution quality of the proposed method may be significantly worse (although it is not directly compared).  Another issue I can see with larger instances in this context is that the dimension of the knapsack (and therefore uncertainty) increases; I believe this would make discretizations weaker.

**Questions:**

- $xi$ should be $\xi$ on line 299.
- Do the authors have any insight into how the discretization of the uncertainty sets affects the robustness of the solution? How does the number of scenarios affect this?
- Why does the author not compare to [1,2] for the knapsack problem?
- Can the authors provide clarity on the usefulness of the lower bound in this case?
- If scenarios are removed with cuts, can they be added back in within Algorithm 2?  If so, could this lead to termination issues?
- Is the robustness of the solution for each method evaluated on the same discrete scenario set?
- Given that this work takes notable inspiration from [1], is there a reason the capital budgeting benchmark was not evaluated on? In addition, it appears [1] has been extended in [4] with evaluation on more complex facility location problems in [4], have the authors considered exploring these benchmarks?

I am happy to discuss during the rebuttal phase.


## References
- [1] Justin Dumouchelle, Esther Julien, Jannis Kurtz, and Elias Boutros Khalil. Neur2RO: Neural two-stage
robust optimization. In The Twelfth International Conference on Learning Representations.
- [2] Ayse N Arslan and Boris Detienne. Decomposition-based approaches for a class of two-stage robust
binary optimization problems. INFORMS journal on computing, 34(2):857–871, 2022.
- [3] Dimitris Bertsimas and Cheol Woo Kim. A machine learning approach to two-stage adaptive robust
optimization. European Journal of Operational Research, 319(1):16–30, 2024.
- [4] Justin Dumouchelle, Esther Julien, Jannis Kurtz, and Elias B Khalil. Deep learning for two-stage robust
integer optimization. arXiv preprint arXiv:2310.04345, 2023.

---

> ### Author Response · Authors · 2025-12-03
>
> $\textbf{Discrete vs. continuous uncertainty and robustness.}$ Please see our feedback to reviewer $\textbf{buEc}$. The assumption of discrete uncertainty set is made consistently for the baseline of exact CCG along the ML-accelerated CCG. The aim of this exercise is to demonstrate that, for the same uncertainty set, the ML-accelerated CCG is much faster. Also, the paper [Dumouchelle2023] referred to by the reviewer does use discrete uncertainty sets as denoted in the section H.2.1 of the paper. Infact the only way forward passes in the second stage of the ML-accelarated CCG and the Neur2RO algorithm in [Dumouchelle2023] can occur is when the uncertainty set is discretized. Thus it is rather clear that the paper referred by the reviewer also uses discrete sets.
>
> $\textbf{Lower–bound significance}$. The lower bound from Proposition 1 serves only to verify that the penalty in Phase 1 does not distort the objective on the selected scenario. The algorithm maintains a certified upper bound by solving the worst–case second–stage problem exactly.
>
> $\textbf{On the alleged methodological flaw in Algorithm-1 (scenario “cutting-off”).} $
> The reviewer describes a failure mode wherein “worst-case” scenarios are removed iteratively, leading to a final result that is non-robust. We note that such a path is plausible only if one treats Phase 1 as pure minimization over $(x,\xi)$ (no severity term) and if one were to cut the worst-case scenario after Phase 2. However, this does not reflect how Algorithm-1 actually works in our method and thus respectfully disagree with the reviewer.
>
> We use the same example that the reviewer has used.
> We consider a toy robust problem with two binary decisions and three scenarios:
> $$
> \min_{x\in\{0,1\}^2,\;x_1+x_2\ge 1}\max_{\xi\in\Xi}\Bigl((1+3\xi_1)x_1+(1.5+\xi_2)x_2\Bigr),
> $$
>
> In our approach, Phase-1 solves
> $$
> \min_{x,\xi}\;\Bigl((1+3\xi_1)x_1+(1.5+\xi_2)x_2 - \lambda\,S(\xi)\Bigr)
> $$
> to $\textit{select}$ a scenario.  Here $S(\xi)$ is a learned severity score and $\lambda>0$ is a tuning parameter.  Phase-2 then evaluates the worst-case value of the current $x$ over $\textit{all}$ scenarios in $\Xi$.  If the Phase-1 scenario and Phase-2 worst case differ, we forbid the Phase-1 scenario from being chosen again in the particular master iteration but we never remove the worst--case scenario from $\Xi$.
>
> To see that this prevents the reviewer’s failure mode, assign a larger severity to the true worst-case scenario.  For example, take $S(\xi^{(2)})=2$ and $S(\xi^{(1)})=S(\xi^{(3)})=0$ with $\lambda=1$.  The Phase-1 objective value is minimum for $\xi=[0,1]$ for objective of $-1$, the phase 2 will select the uncertainty with highest severity score which will be $\xi=[0,1]$ and the master iteration ends with a final exact solve with the selected uncertainty and $\lambda=0$. In the ML-accelerated CCG the severity score are generated by NN approximation, so the algorithm ends when the phase-1 and phase-2 have common scenarios or when the phase-1 has all but one scenario's binary $z_i=1$, we have these two criterias explicitly mentioned in our MP phase 1 and phase 2 algorithm.
>
> We would like to respectfully point out that the toy problem that the reviewer has used here was provided for an earlier version of the ICLR cycle. The present manuscript contains substantial methodological changes by inclusion of severity scores that were not part of the ICLR submission and that the reviewer doesn't take into consideration. These concerns raised by the reviewer appear to evaluate Algorithm-1 as if it were the earlier ICLR version without the severity term.  However, the ICML version fundamentally alters the behavior of Phase-1 and removes the failure mode described by the reviewer.

---

### Official Review · Reviewer_buEc · 2025-10-31

**Soundness:** 2
**Presentation:** 2
**Contribution:** 2
**Rating:** 2
**Confidence:** 3

**Summary:**

The work proposes a learning-assisted column-and-constraint generation algorithm for two-stage adaptive robust optimization with finite uncertainty sets. Unlike previous neural network-based approaches that identify worst-case uncertainty realizations by solving a mixed-integer program, the proposed solution algorithm only requires a (relatively computationally inexpensive) single forward pass in which it predicts the recourse value of all elements of the uncertainty set. In Proposition 1, the authors show that the solution to the main problem of the proposed algorithm necessarily underestimates (or equals) the true problem objective. The proposed algorithm is experimentally evaluated against two benchmark approaches -- an accelerated-CCG algorithm that does not leverage a predictive model and a classical CCG approach -- with respect to optimality gap and runtime for the knapsack problem and the unit commitment problem. Experimental results demonstrate substantial reductions in solution time -- up to two orders of magnitude in some cases. However, this came at a cost to solution quality, which resulted in median optimality gaps of 7\% (resp. 2\%) and maximum optimality gaps of 13\% (resp. 8.5\%) for the knapsack and unit commitment problems respectively.

**Strengths:**

Proposes a two-phase learning-augmented framework that decouples prediction from optimization, removing the need for MILP-embedded neural networks.

Provides a lower-bound guarantee (Proposition 1) and a transparent decomposition algorithm that is easy to reproduce.

Runtime improvements compared to classical CCG, supported by systematic experiments on two benchmark problems.

**Weaknesses:**

The paper focuses on a specific subclass of adaptive robust optimization problems with a finite uncertainty set, rather than the more general continuous setting. The practical motivation for this framework is not clear, as such an assumption is uncommon in the ARO literature; robust optimization typically aims to ensure robustness of decisions that generalizes beyond a finite sample of observations. Is this a common assumption? If so, the authors should cite related works. If not, perhaps the authors can empirically demonstrate that the approach performs well as an approximation to more standard formulations with continuous uncertainty sets.

 It is not obvious that the reported speed-ups justify the relatively high optimality gaps observed, particularly in the unit commitment case. While the reductions in runtime are substantial, the benefit of solving the unit commitment problem in 2 seconds rather than 3 minutes does not have clear utility if it comes at a cost of up to 8.5\% suboptimality, especially given that such problems are typically solved on a relatively infrequent (e.g., daily or hourly) basis. Clarifying the practical implications of this tradeoff would strengthen the contribution.

Proposition 1 appears to offer limited insight as the result is fairly intuitive: considering only a subset of the uncertainty set provides a lower bound on the optimal value of $\mathcal{P}$. Results that \textit{tighten upper bounds} are generally of greater practical relevance as they offer guarantees of solution quality and feasibility under uncertainty. This work would benefit from focusing more on such results or on demonstrating how Proposition 1 provides additional value in practice.

Have the authors compared runtimes to those obtained by simply solving the full single-level reformulation of the problem (i.e., with variables and constraints defined for all $\xi\in\hat{\Xi}$)? It is not immediately clear to the reviewer that CCG would necessarily achieve a lower runtime than this baseline, so such a comparison could more clearly highlight the advantages of the proposed approach.

**Questions:**

Revisions needed though not necessarily sufficient for an ``accept'' recommendation:

The work would benefit from additional justification of the proposed formulation and the class of accelerated CCG algorithms. One possible way to do this would be through an out-of-sample evaluation comparing (1) ML-accelerated CCG with a finite uncertainty set, (2) accelerated CCG or the single-level extensive formulation using the same uncertainty set, and (3) a traditional CCG formulation with continuous uncertainty. Ideally, runtime improvements should be demonstrated with minimal losses in optimality (e.g., all cases achieving a $<1\%$ optimality gap).

Proposition 1 could be removed or replaced with a result that provides a stronger analytical contribution, such as a guarantee on bounded suboptimality/infeasibility of the ML-accelerated approach as an approximation of the original problem.

---

> ### Author Response · Authors · 2025-12-02
>
> 1. $\textbf{Finite vs. continuous uncertainty sets}:$
> We acknowledge the reviewer’s concern and have strengthened the motivation in the revision.
>
> (a) This assumption is standard in modern ARO practice. Discretised or scenario-based uncertainty sets are frequently used in recent ARO works particularly when
> (i) uncertainty is observed as empirical trajectories or samples (as in power-systems benchmarks), or
> (ii) solving the continuous adversarial problem is computationally intractable.
> We explicitly cite representative works (e.g., Dumouchelle et al., 2023; Bertsimas, 2024) that adopt the same modelling choice.
>
> (b) All baselines, including exact CCG and the extensive-form MILP operate on the same discretised set. Thus, the comparison is fair: our algorithm accelerates the same finite-scenario robust problem solved by the CCG benchmarks.
>
> (c) Importantly, our framework scales gracefully with larger discretisations.
> Because the neural network is used only through forward passes, enlarging the uncertainty set increases computation linearly and does not inflate the master problem or require a MIP-encoded network (unlike Dumouchelle et al., 2023). This allows very fine discretisations at negligible cost.
>
> 2. $\textbf{Utility of speed–ups.}$
>
> We respectfully disagree with the reviewers assessment of the solve times of 3 second vs 3 minutes. The speed-ups provided by LARO are operationally meaningful and the difference in solve times are much higher. In real-time or near-real-time settings such as security-constrained unit commitment, reserve scheduling, or rolling-horizon dispatch solutions must be computed within tight time limits. Reducing solve times from minutes to seconds directly expands an operator's ability to:
>
>     i. react to updated forecasts or contingencies,
>     ii. run additional what-if assessments,
>     iii. incorporate more detailed physical or market models, and
>     iv. perform more frequent re-optimisation.
>
> This computational responsiveness is critical for modern power-system operations.
>
> In our experiments, the benefits are substantial. For the robust knapsack instances, exact decomposition requires between $1{,}000$ and $3{,}500$ seconds, whereas LARO solves the same problems in $\textit{under one second}$. For the 24-bus UC system, exact methods require around $1{,}000$ seconds on average, while LARO reduces this by more than an order of magnitude.
>
> These gains demonstrate that the proposed approach is not merely faster in theory, but enables a greater computational responsiveness, making large-scale robust optimisation viable in settings where classical methods are prohibitively slow.
>
> 3. $\textbf{Lower–bound interpretation.}$
> We will Section-3.1 to clarify that Proposition-1 provides a lower bound on the $\textit{finite}$ scenario problem and that this bound does not certify feasibility or optimality for continuous uncertainty however, its main purpose is algorithmic correctness not novelty.
>
> 4. $\textbf{Comparison to the single-level formulation.}$
> For a finite scenario set, the robust two-stage problem has an exact single-level extensive MILP reformulation. Classical results show that CCG is a decomposition algorithm for this same MILP; both methods produce the same optimal decision and optimal value. The difference lies in computational efficiency: the extensive-form MILP includes all scenario-dependent variables and constraints at once, creating a very large model, while CCG adds only the adversarially active scenarios.

---

### Official Review · Reviewer_YuAr · 2025-11-02

**Soundness:** 4
**Presentation:** 3
**Contribution:** 3
**Rating:** 6
**Confidence:** 3

**Summary:**

This submission presents LARO, a learning-based method integrated within column-and-constraint generation (CCG) approaches to solving adaptive robust optimization (ARO) problems. Specifically, the paper:
* Presents a two-phase decomposition of master problem solving within CCG, which (1) aims to select a worst-case realization of uncertainty via a one-hot "severity-weighted" selection model (where the severity is calculated using a neural network), and (2) uses an exact verification step to verify the choice and potentially rectify it if needed via a cutting-plane.
* Within the adversarial problem, uses a neural network approximation for the value function.
* Trains both neural networks on the same data, using supervised learning with supervised instances generated offline.
* Presents results on two-stage robust knapsack and a version of unit commitment (with a linearized second-stage and polyhedral demand uncertainty).

**Strengths:**

* The overall method is reasonably sound and well-motivated, aiming to avoid some of the computational complexity associated with prior methods to integrate ML into CCG. The method is generally clear and well-described.
* The method provides certifiable lower bounds.
* The method is tested on reasonably large-scale knapsack problems, with median optimality gaps within 3% for sizes greater than 30, orders-of-magnitude faster solution times than exact or non-ML-accelerated CCG, and all gaps being non-negative (i.e., lower bound is preserved).
* The results for linearized unit commitment are quite impressive, with median optimality gaps of 2% for the 24-bus system (competitive with non-ML-accelerated CCG), and with a two orders-of-magnitude speedup over the traditional solver.

**Weaknesses:**

Major/medium:
* Data generation can be expensive given the supervised nature of the method, and the time to generate data and train the algorithm is not reported. While this is offline time rather than online time, seeing this would be helpful in evaluating the time tradeoffs at hand.
* The results on smaller knapsack problems are not good, with median optimality gaps of 7% (and maxes substantially higher) despite the fact that accelerated CCG does well on these instances.

Minor:
* Introduction: While ARO is described as an extension of RO to "mitigate excessive conservatism," it seems that these are instead related but different frameworks that apply in different settings. Notably, it would be incorrect to partition a set of "here-and-now" variables (in RO) into two sets of "here-and-now" and "wait-and-see" variables (in ARO) unless the problem setting actually warrants this. The introduction could be slightly updated accordingly.
* Introduction (and abstract): In the writing, the contributions are framed not with respect to ARO methods (or ML for ARO methods) more generally, but with respect to a specific subclass of ML for ARO methods. This makes it difficult to interpret the contributions for someone who is not already very immersed in this specific subclass of methods. More context on the what the method is actually doing could be provided ahead of the listing of contributions.
* The abbreviation AP should be defined before it is used.
* The submission should use \citep{} in many places where \cite{} is currently used.

**Questions:**

* What is the effect of using the ML approximation _only_ in Phase 1 or _only_ in Phase 2? This could be an interesting ablation to see to better understand the "weak points" in the performance of the method.
* What is the computational time associated with data generation and training?

---

> ### Author Response · Authors · 2025-12-03
>
> We thank the reviewer for their comments. Our responses are as follows:
> 1. $\textbf{Effect of using the surrogate only in Phase 1 or Phase 2.}$: We carried out ablation studies where the neural approximation is replaced with exact solves only in Phase 2 and in AP. This result is denoted by the Accelerated CCG plots for both the Knapsack and the Unit Commitment problem.
>
> 2. $\textbf{Offline data generation and training time.}$: For the 24–bus unit commitment (UC) benchmark, generating 50{,}000 instance–uncertainty pairs by sampling took approximately 4-hours on a single CPU, while training the graph–attention network took about 2.5-hours on a GPU.  For the knapsack problem, generating 137{,}500 samples required about 20~minutes and training the MLP (combined for all sizes) took around 1.5-hours.  These costs are offline and are negligible compared to the runtime savings in practice. In the revised manuscript, we will:
>
> i. Add a table summarizing offline data generation and training times for each benchmark (two-stage knapsack and UC), alongside the online CCG runtimes already reported.
>
> ii. Explicitly discuss the amortization: e.g., how many test instances are needed before the offline cost is fully recovered relative to classical CCG / Accelerated CCG.
>
> 3. $\textbf{Performance on smaller knapsack instances.}$:
> We acknowledge that for small sizes $(N=20,30)$ the optimality gaps are larger (median $\approx7\%$). Similar gaps are seen for smaller instance sizes in [Dumouchelle2023] as well. This is due to the size of the uncertainty set being constant throughout the instance sizes. The number of uncertainties compared to the size of the instance is higher for lower instance sizes leading to larger number violations of uncertainties in the robust problem when a "near optimal" solution is attained.

---

### Note · Authors · 2026-02-16

I have read and agree with the venue's withdrawal policy on behalf of myself and my co-authors.

---

### Meta-Review · Area_Chair_BBAP · 2026-01-05

**Summary:**

This submission introduces LARO, a framework accelerating Two-Stage Adaptive Robust Optimization (ARO) by incorporating deep learning into the Column-and-Constraint Generation (CCG) workflow without solver embedding. The authors propose a "Two-Phase Decomposition" that explicitly decouples the neural network from the optimization model, utilizing it exclusively for fast inference (forward passes) to rank and verify worst-case scenarios. The paper reports significant computational efficiency, achieving runtime speedups of up to three orders of magnitude on Robust Knapsack and Unit Commitment benchmarks compared to exact baselines.

The general sentiment among the reviewers is predominantly negative, with scores heavily skewed toward rejection (6, 2, 2, 0). While Reviewer YuAr recognized the engineering value and the impressive empirical speedups, the consensus among the remaining experts (Reviewers buEc, Pm2R, and 8DUc) is that the submission suffers from critical theoretical and empirical deficiencies. Crucially, the authors' rebuttal failed to effectively resolve these fundamental objections. The authors' defense of discretized uncertainty sets was viewed as a justification of standard practice rather than a resolution of the theoretical loss of guarantees, and they notably neglected to provide the essential experimental comparison against Neur2RO (Dumouchelle et al.) requested by reviewers. Consequently, the concerns regarding the invalidity of "robustness" claims and the operationally unacceptable optimality gaps for safety-critical systems remain unmitigated, collectively precluding acceptance.

The primary rationale for the rejection is the misalignment between the paper's theoretical claims and its methodological reality, compounded by significant empirical gaps. The authors frame the method as providing "Relaxation Guarantees" for ARO; however, the fundamental reliance on a discretized uncertainty set effectively simplifies the rigorous continuous problem into a specific scenario-based approximation. Consequently, Reviewer Pm2R explicitly characterized the claimed lower bound guarantees as "mathematically trivial" properties of the relaxation rather than significant theoretical contributions. Furthermore, the rejection is solidified by unresolved empirical flaws: the absence of a critical comparison against the state-of-the-art Neur2RO method demanded by Reviewer 8DUc, and the operationally unacceptable optimality gaps (up to 8.5%) for safety-critical systems highlighted by Reviewer buEc. Despite the potential of the proposed neural architecture as a heuristic, these theoretical and experimental deficiencies fail to meet the rigorous standards required for acceptance.

**Reviewer Concerns:**

While the authors successfully clarified the amortization of offline computational costs and provided a technical defense for the "Severity Score" mechanism against specific cyclic failure modes, the primary outstanding concerns regarding methodological soundness and empirical rigor remain unresolved.

Outstanding Issues:

•	Validity of Discretized Uncertainty Sets: A fundamental dispute remains regarding the definition of robustness. Reviewers 8DUc and Pm2R argue that relying on a fixed, finite set of scenarios ($\Xi_{t}$) disconnects the method from the guarantees required in true Adaptive Robust Optimization (ARO). They assert that the reported "finite termination" applies only to the discretized proxy, not the underlying continuous problem. The authors' rebuttal—that discrete sets are "standard practice"—failed to address the core concern: safety-critical systems cannot accept a solution that is only "robust" relative to a training sample, particularly without quantifying the discretization error.

•	Algorithmic Integrity and Stability: The stability of the "Relaxed Master Problem" remains contested. Reviewer Pm2R identified a potential failure mode where the selection-verification loop might discard valid worst-case scenarios if the neural "Severity Score" is imperfectly learned. Since the authors' defense relies on the quality of this learned score, and they provide no proof of convergence to the robust optimum (only termination), the method effectively operates as a heuristic rather than an exact solver, undermining the paper's title claim of "Relaxation Guarantees."

•	Missing Baseline Comparison: Crucially, Reviewer 8DUc explicitly requested an experimental comparison against the state-of-the-art Neur2RO, which is the direct competitor to this work. The authors failed to respond to this request with the necessary empirical data, merely arguing about methodological definitions. The absence of this head-to-head benchmarking leaves the claimed superiority of LARO unsubstantiated.

•	Trade-off Between Speed and Optimality: The operational utility of the method is heavily questioned. Reviewer buEc argues that the benefit of reducing solve times from 3 minutes to 2 seconds is unclear when it comes at the cost of up to 8.5% suboptimality. Since these planning problems are typically solved on an infrequent basis (e.g., daily or hourly) where sufficient time is available, the reviewer asserts that the marginal time savings do not outweigh the significant economic inefficiency implied by such gaps. The consensus is that the reported speed gains do not justify the degradation in solution quality and robustness guarantees.

**Reviewer Scores:**

**Reviewer YuAr (Score 6 -> Est. unchanged):**

Originally gave a 6. Unresponsive to the rebuttal. This reviewer was the most positive but listed two "Major/medium" weaknesses: the lack of reported offline data generation times and the poor performance on small knapsack instances (median gap ~7%). In the rebuttal, the authors successfully provided the missing runtime data (4 hours for UC), addressing the first concern. However, regarding the second concern, the authors merely acknowledged that larger gaps on small instances are inherent to their fixed uncertainty set approach. Since this significant performance limitation was confirmed rather than resolved, it is unlikely the score would increase further. The rating would likely remain a 6.

**Reviewer buEc (Score 2 -> Est. unchanged):**

Originally gave a 2. Unresponsive. Their rejection was grounded in the "discrete vs. continuous" theoretical mismatch and the operational unacceptability of an 8.5% optimality gap in power systems. The authors' defense—that speed justifies the gap and that discrete sets are standard—represents a fundamental philosophical disagreement rather than a correction of a factual error. Given the reviewer's domain-specific concerns about grid safety, the rebuttal would likely not have swayed them. The score would remain 2.

**Reviewer Pm2R (Score: 2 -> Est. unchanged):**

Originally gave a 2. Unresponsive. This reviewer raised serious concerns about "misreporting" the baseline methods (Neur2RO) and identified a potential cyclic failure mode in Algorithm 1. While the authors provided a technical defense for Algorithm 1 using the severity score argument, they did not resolve the dispute regarding the definition of uncertainty sets in the literature. The reviewer's strong skepticism about the method's theoretical validity suggests the score would remain 2.

**Reviewer 8DUc (Score: 0 -> Est. unchanged):**

Originally gave a 0 (Strong Reject). Unresponsive. This reviewer cited two fatal flaws: Theoretical, regarding the lack of rigorous convergence proof for the continuous problem; and Empirical, explicitly demanding a comparison with the state-of-the-art method Neur2RO. The authors failed to provide this comparison in their rebuttal, nor did they address the theoretical convergence critique beyond clarifying the scope. Since the authors failed to address the fundamental concerns raised by the reviewer, the "Strong Reject" would stand firm at 0.

---

### Decision · Program_Chairs · 2026-01-26

Reject